# Are EARLINET and AERONET climatologies consistent? The case of Thessaloniki, Greece

Nikolaos Siomos[1], Dimitris S. Balis[1], Kalliopi A. Voudouri[1], Eleni Giannakaki[2,4], Maria Filioglou[1,2], Vassilis Amiridis[3], Alexandros Papayannis[5], and Konstantinos Fragkos[6]

[1]Laboratory of atmospheric physics, Physics Department, Aristotle University of Thessaloniki, Greece, *Email: nsiomos@physics.auth.gr
[2]Finnish Meteorological Institute, Atmospheric Research Center of Eastern Finland, Kuopio, Finland
[3]Institute for Astronomy, Astrophysics, Space Applications and Remote Sensing, National Observatory of Athens, Athens, Greece
[4]Department of Environmental Physics and Meteorology, Faculty of Physics, University of Athens, Greece
[5]National Technical University of Athens, Physics Department, Laser Remote Sensing Laboratory, Athens, Greece
[6]National Institute of R&D in Optoelectronics, Magurele, Romania

*Correspondence to:* N. Siomos (nsiomos@physics.auth.gr)

**Abstract.** In this study we investigate the climatological behavior of the aerosol optical properties over Thessaloniki during the years 2003-2017. For this purpose, measurements of two independent instruments, a lidar and a sunphotometer, were used. These two instruments represent two individual networks, the European Lidar Aerosol Network (EARLINET) and the Aerosol Robotic Network (AERONET). They include different measurement schedules. Fourteen years of lidar and sunphotometer measurements were analyzed, independendly of each other, in order to obtain the annual cycles and trends of various optical and geometrical aerosol properties in the boundary layer, in the free troposphere and for the whole atmospheric column. The analysis resulted in consistent statistically significant and decreasing AOD 355nm trends of -23.2% and -22.3% per decade in the study period over Thessaloniki for the EARLINET and the AERONET datasets respectively. Therefore, the analysis indicates that the EARLINET sampling schedule can be quite effective in producing data that can be applied to long-term climatological studies. It is also shown that the observed decreasing trend is mainly attributed to changes in the aerosol load inside the boundary layer. Seasonal profiles of the most dominant aerosol mixture types observed over Thessaloniki have been generated from the lidar data. The higher values of the vertical-resolved extinction coefficient at 355nm appear in summer, while the lower ones appear in winter. The dust component is more dominant in the free troposphere than in the boundary layer during summer. The biomass burning layers tend to arrive in the free troposphere during spring and summer. This kind of information can be quite useful for applications that require a priori aerosol profiles. For instance, they can be utilized in models that require aerosol climatological data as input, in the development of algorithms for satellite products, and also in passive remote sensing techniques that require knowledge of the aerosol vertical distribution.

# 1 Introduction

The atmospheric aerosol load typically shows a significant spatial and temporal variability within the lower atmosphere (e.g., Hamill et al., 2016). This is related both to the plethora of aerosol emission sources near the ground and to the variable weather conditions that appear in the troposphere. Since transport is driven by the atmospheric conditions, the aerosol properties over a given location are expected to follow annual and climatological patterns just as the wind does (e.g., Takemura et al., 2002). Similar patterns can be observed in the emission sources as well (e.g., Stefan et al., 2013). As a matter of fact, a lot of human activities, that result to the emission of anthropogenic aerosols, exhibit annual cycles (e.g., Yiquan et al., 2015). This is also true for the natural emissions that are usually driven by the weather conditions (e.g., Israelevich et al., 2012). The knowledge of the climatological behavior of particles in the troposphere can be utilized in many different ways. Its applications can range from purely scientific, such as the validation of aerosol transportation and air quality models (e.g., Binietoglou et al., 2015; Siomos et al., 2017) and satellite instruments (e.g., Balis et al., 2016) to civil oriented, for example the impact of the aerosol load on human health (e.g., Mauderly and Chow, 2008; Löndahl et al., 2010), airfare safety (e.g., Brenot et al., 2014) and agriculture (e.g., Gerstl and Zardecki, 1982).

In order to conduct a climatology study, long-term scheduled measurements are required. In situ techniques focus on measurements of the aerosol properties close the ground. It is both challenging and costly to acquire those measurements in high altitudes (i.e. mounted on airplanes and unmanned aerial vehicles), especially on a routine basis. For those reasons, the application of remote sensing techniques from ground based instruments is usually preferred. Lidar systems are ideal when the vertical distribution is being investigated (e.g., Klett, 1981; She et al., 1992; Ansmann et al., 1992; Welton et al., 2001; Hirsikko et al., 2013). Passive remote sensing instruments are also broadly used in order to examine the columnar aerosol properties (e.g., Dubovik and King, 2000; Hönninger et al., 2004; Schneider et al., 2008; Herman et al., 2017; López-Solano et al., 2017).

Previous climatological studies using Raman lidar measurements at Thessaloniki were conducted by Amiridis et al. (2005) and Giannakaki et al. (2010) covering the periods 2001-2004 and 2001-2007 respectively. These studies focus on the seasonal variability of various aerosol optical properties inside the planetary boundary layer and in the free troposphere, separately for the predominant aerosol mixtures. For example, Amiridis et al. (2005) have found a seasonal pattern in the columnar AOD, with higher values occurring mainly in early spring and late summer due to an enhanced free tropospheric contribution, while Giannakaki et al. (2010) observed larger optical depth values for Saharan dust and smoke particles. However, the limited number of years did not permit the calculation of long term trends. On the other hand, Kazadzis et al. (2007) and Fountoulakis et al. (2016) analyzed longer datasets, based on spectral irradiance measurements for Thessaloniki, that allowed them to investigate the long-term variability and the annual cycles of the aerosol optical depth in the UV for Thessaloniki. They used retrievals of AOD from two different Brewer spectrophotometers in the periods 1997-2005 and 1994-2006 respectively. For instance, Kazadzis et al. (2007) detected a seasonal variation in the monthly means of AOD at 340nm with maximum optical depth values in the summer months and minimum in wintertime, while Fountoulakis et al. (2016) detected an AOD at 320nm trend of -0.09 $\pm$ 0.01 per decade. In their case, however, it was not possible to provide information on the aerosol vertical distribution due to the nature of their instrumentation.

In this study we have investigated the climatological behavior of the aerosol optical and geometrical properties over Thessaloniki during the period 1st of June 2003 to 31st of May 2017, which, hereafter, will be referred to as "period 2003-2017". We have used the measurements of two independent datasets that represent two individual networks with different measurement schedules and techniques.

The first dataset includes measurements performed with a Raman lidar in Thessaloniki, Greece (40.63$^o$ N, 22.96$^o$ E). This instrument is part of the European Aerosol Lidar Network (EARLINET). The EARLINET schedule for climatological measurements is adopted (e.g., Giannakaki et al., 2010) and measurements are systematically performed every Monday morning preferably close to 12 UTC, and every Monday and Thursday evening, preferably after sunset, resulting in 302 days with measurements. After the CALIPSO mission in 2006, lidar measurements have also been performed during the CALIPSO over-

passes (Winker et al., 2009; Pappalardo et al., 2010), resulting in 73 additional days with lidar data. Finally, depending on the station's needs, measurements are performed during special events, resulting to 143 additional days of data. The full dataset includes 518 days when at least one lidar profile is available. The second dataset includes data measured with a CIMEL sun-photometer that is part of the Aerosol Robotic Network (AERONET). Measurements are automatically performed every 15 minutes or less, depending on the sun's zenith angle (Holben et al., 1998; Dubovik and King, 2000). By using these data, the

long-term variability, the annual cycles, and trends of various optical, and geometrical properties have been examined. Furthermore, we have separately investigated the climatological behavior of aerosols in the planetary boundary layer (PBL) and in the free troposphere (FT). Taking into account the different sampling rate of the two datasets and the different measurement techniques, the aim of our study was to ultimately reach a more solid conclusion regarding the capability of the two datasets to produce consistent climatological patterns, when analyzed independently of each other. It is not in our intent to perform a point

by point comparison of coincident in time measurements between the two techniques. However, the uncertainties involved in producing the climatological datasets are discussed in section 4.5.

## 2    Instrumentation and tools

### 2.1    The lidar system

The setup of the lidar system is discussed in this section. It belongs to the Laboratory of Atmospheric Physics that is located

in the Physics department of the Aristotle university of Thessaloniki (40.63$^o$ N, 22.96$^o$ E) at an elevation of 50 m. The first (1064nm), second (532nm) and third harmonic (355nm) frequency of a compact, pulsed Nd:YAG laser are emitted with a 10 Hz repetition rate (more technical details can be found on (Amiridis et al., 2005)). The radiation from the atmospheric backscattering of the laser beam is collected with a 500 mm diameter telescope. The lidar has been part of EARLINET (Schneider et al., 2000; Pappalardo et al., 2014) since 2000. The original setup of the Raman lidar in 2000 included two elastic

channels at 355nm and 532nm and a Raman channel at 387nm (Amiridis et al., 2005). More channels were added later on. An additional Raman channel at 607nm was added in 2008. Another elastic channel at 1064nm plus one parallel and one cross polarization channel at 532nm were added in 2012 (Siomos et al., 2017). The final products, which derived from the raw lidar data processing (see section 3.2) are the aerosol backscatter coefficient at 355nm, 532nm and 1064nm and the aerosol

extinction coefficient at 355nm and at 532nm. The dataset included in this study covers the period 2003-2017 in order to be chronologically consistent with the sunphotometer dataset (see section 2.2). All of the aforementioned products are publicly available in the EARLINET database (https://www.earlinet.org).

## 2.2 Lidar overlap function

A common source of uncertainty when dealing with lidar data is the system's overlap function that determines the altitude above which a profile contains trustworthy values. For simplicity we will refer to this altitude as "starting height" in the manuscript. In our analysis, if a correction is not available for the system's overlap, the starting height is set to the full overlap height. This is true for all our daytime elastic backscatter profiles and the night-time elastic backscatter 532nm profiles prior to 2008. The starting height is below 1.5 km for 86% of those profiles. The Raman extinction profiles are much more sensitive to the overlap

effect (see section 3.2). The method of Wandinger and Ansmann (2002) is applied if Raman profiles are available and the overlap function is calculated and applied individually per Raman case. The correction is also applied to the night-time elastic backscatter at 1064nm that became available in 2012. The calculated overlap function can be trusted for values greater than 0.7 (Amiridis et al., 2005). In those profiles, the starting height is set to the altitude where the overlap equals 0.7, resulting in values below 1.5km for 90% of the overlap corrected profiles. For the calculation of the columnar properties, a constant profile

is assumed from the starting height to the ground. This introduces uncertainties in the calculation of the AOD. The impact of these uncertainties in the climatological analysis will be discussed in section 4.5.

## 2.3 The sunphotometer

The CIMEL multiband sun-sky photometer was installed in Thessaloniki in 2003 as part of the AERONET Global Network. It is located at the same altitude as the lidar system. Their distance is less than 50 m. It performs direct solar irradiance and sky

radiance measurements at 340, 380, 440, 500, 670, 870, and 1020nm automatically during the day. The AERONET inversion algorithms (Dubovik and King, 2000; Dubovik et al., 2006) are applied automatically to the raw data. The products are publicly available online (https://aeronet.gsfc.nasa.gov). The level 2.0 Version 3 aerosol optical depth values (AOD) at 440, 675, 870 and 1020nm in the period 2003-2017 were used in this study in order to take advantage of the longer timeseries since the 340nm and 380nm channels were added later, in 2005 and were also missing for the period 2008-2011 due to changes of the

instrument. A conversion technique is applied in order to calculate the sunphotometer AOD in lidar-compatible wavelengths. It is discussed in section 3. Details on the instrument and the AERONET infrastructure are included in (Holben et al., 1998).

## 3    Methodology

The pre-prossessing required in order to obtain the final climatological products is discussed in this section. The full dataset is applied for the calculation of the aerosol geometrical properties. The lidar dataset applied for the calculation of the aerosol

optical properties is a subset that includes the night-time aerosol extinction profiles at 355nm and the corresponding aerosol

backscatter profiles at 355nm and 532nm (section 2.1), while the sunphotometer dataset contains AOD data at 440, 675, 870, and 1020nm (section 2.2).

Further processing is required in order to get some structural elements from the lidar profiles. These structural elements are often referred to as geometrical properties. In our analysis, we have calculated the boundary layer height and the first major lofted layer base, top and center of mass height. With this information the AOD within the PBL and the FT can be distinguished. The aerosol optical depth (AOD) at 355nm is calculated from the integration of the lidar extinction profiles. The integrated backscatter coefficients at 355nm and 532nm are also obtained from the EARLINET dataset. Finally, some intensive optical products that are characteristic of the aerosol type and derive from the backscatter and the extinction profiles have been calculated. This includes the extinction to backscatter ratio, often referred to as the lidar ratio, at 355nm and the backscatter-related Angstrom exponent in the spectral region 355-532nm. The former depends mostly on the absorption and scattering aerosol properties, while the latter depends mainly on the aerosol size distribution. The analysis covers both the profile and the columnar versions of these products.

An overview of the EARLINET dataset is provided in section 3.2. The pre-processing required in order to calculate the geometrical optical properties from the lidar profiles are described in sections 3.3 and 3.4 respectively.

## 3.1 Sunphotometer pre-processing

It is necessary to make the sunphotometer optical depth compatible with the lidar optical depth at 355nm. An extrapolation method is applied (Soni et al., 2011) in order to obtain the AOD at 355nm from the sunphotometer data. This method assumes a 2nd order polynomial relationship for the logarithm of the AOD in the spectral region 340-1020nm. The constant Angstrom approach is equivalent to a linear fit to the logarithm of the AOD, instead. The 2nd order polynomial is calculated by fitting the sunphotometer AOD values at 440, 675, 870, and 1020nm in a logarithmic scale. Cases with too low AOD 440nm values, below 0.05, and cases where the polynomial is ill-fitted are excluded. The AOD 355nm is then extrapolated from the polynomial, assuming that it is also valid in the UV region. The validity of the conversion is tested with the sunphotometer AOD at 340nm for the periods when both were available. In figure 1, the extrapolated AOD at 340nm, using both the 2nd order polynomial and the linear fit methods, is compared with the measured AOD at 340nm. The 'linear' method tends to systematically produce higher extrapolated AOD, especially for the cases with high AOD. This behavior is also present in the 'polynomial' approach, but it is much less pronounced. In this case, the absolute bias is below 0.035 for 90% of the cases. The sunphotometer uncertainty is 0.02 and should be even higher for the UV (Kazadzis et al., 2016). Consequently, this conversion ensures that the error introduced by the AOD extrapolation is typically close to the sun-photometer uncertainty.

## 3.2 Dataset overview

Many techniques and methods have been developed for the lidar signal pre-processing and inversions (e.g., Klett, 1981; Fernald, 1984; Ansmann et al., 1992; Lopatin et al., 2013; Chaikovsky et al., 2016). In order to ensure qualitative and consistent data processing within the EARLINET network, algorithm intercomparison campaigns have been organized (Matthias et al., 2004; Pappalardo et al., 2004; Böckmann et al., 2004). These campaigns aimed to establish the standard methods that can be utilized

by all the stations. Additionally, some quality standards have been established, in order to make the lidar products of the different systems comparable and to be able to provide quality-assured data sets of network products (Freudenthaler et al., 2018).

Concerning the timeseries under study, two different methods of processing are applied depending on the type of measurement. During the day, the data acquisition is limited to the signals that occur from the elastic scattering of the laser beam by the air molecules and the atmospheric aerosol. The Klett-Fernald-Sasano (KFS) inversion is applied (Klett, 1981; Fernald, 1984; Sasano and Nakane, 1984) and the backscatter coefficient profiles are produced. A constant a-priori climatological value of the lidar ratio has to be assumed in this method. The resulting uncertainties are discussed in depth by Böckmann et al. (2004) and can be as high as 50% if there is no information about the actual lidar ratio.

In the night, the vibrational Raman bands of the atmospheric nitrogen at 387nm and 607nm can be recorded. In this case, the Raman inversion (Ansmann et al., 1992) is applied. It allows the calculation of both the extinction and the backscatter profiles without any assumption regarding lidar ratio. Nevertheless, a constant a-priori value of the Angstrom exponent between the elastic and the Raman wavelength has to be assumed. The relative error introduced should be less than 4% (Ansmann et al., 1992). The technique described in Wandinger and Ansmann (2002) allows the calculation of the lidar system's overlap function from Raman measurements. The correction is applied individually to each Raman measurement. This is particularly important for the calculation of the extinction profiles. They are calculated using the inelastic signal height derivative (Ansmann et al., 1992). As a result, they are very sensitive to the system's overlap function.

A time versus height cross section of the aerosol extinction coefficient at 355nm for the period 2003-2017 is presented in figure 2. It gives an overview of the availability of the lidar measurements. The monthly mean values are produced using every available measurement. The long gaps in the years 2008 and 2011 of the timeseries are attributed to system upgrades. Some missing months also occur, especially during winter, when the weather conditions are not favorable for lidar measurements. The aerosol load seems to be significant only below 4km in most cases. The highest extinction values are typically observed closer to the ground, as expected. This is attributed to the mixing mechanisms that take place near the surface. Elevated layers can also be observed, especially in the summer months. Geometrical features that are representative of the vertical distribution of the aerosol load can be obtained from the lidar profiles. In section 3.3 we discuss the algorithmic processes that are required in order to extract those features.

### 3.3 Geometrical properties

The aerosol geometrical properties carry information about the structure of lidar profiles. Examples are the boundary layer height and the boundaries of the lofted layers. They can be obtained from any lidar profile. As a result, the full lidar dataset presented in section 2.1 has been applied for the calculations. Some lidar products, however, are more accurate to use than others. For example, the longer wavelengths typically magnify the differences in the vertical distribution of the aerosol load, resulting in layers that are easier to identify. Furthermore, the Raman inversion always results in profiles that are less structured for the extinction coefficients than the backscatter coefficients. This is the reason why we prioritize them in order to produce geometrical properties. The product with the highest potential to magnify the layer structure available is selected for each mea-

surement. More specifically, the backscatter products are prioritized over the extinction products and the longer wavelengths over the shorter ones.

### 3.3.1 Boundary layer height detection

Many methods have been proposed for the calculation of the PBL height from lidar data (e.g., Flamant et al., 1997; Menut
et al., 1999; Brooks, 2003; Tomasi and Perrone, 2006; Haeffelin et al., 2012; Milroy et al., 2012; Bravo-Aranda et al., 2016). Our analysis is based on the method of Baars et al. (2008) that applies the wavelet covariance transform (WCT) to the raw lidar data in order to extract geometrical features such as the PBL height and the cloud boundaries. In our case, we want to apply this method to the database products instead. The WCT transformation has also been applied successfully in the past on other lidar products. Siomos et al. (2017), for example, use an adaptation of the WCT method and calculate the geometrical features
from the aerosol concentration profiles. The transform is provided by equation 1.

$$W(\alpha, z) = \frac{1}{\alpha} \left( \int\limits_{z-\frac{\alpha}{2}}^{z} F(z') dz' - \int\limits_{z}^{z+\frac{\alpha}{2}} F(z') dz' \right) \tag{1}$$

where F is the product profile which the transform is being applied to, W is the result of the transformation, z and z' is the altitude and $\alpha$ is the dilation. A dilation of 0.4 km is used for the PBL height calculations, similar to Baars et al. (2008). Additionally, an upper limit is necessary so that the top of elevated layers is not misidentified as the PBL (Baars et al., 2008).
We use an upper limit of 4.2 km to be consistent with previous studies over the area (Georgoulias et al., 2009).

The boundary layer is evolving during the day and reaches its maximum height at 12 Local Solar Time. Consequently, as far as the daytime measurements are concerned, we preferred to use only measurements performed between 10 and 13 UTC. After sunset, the boundary layer collapses fast and the stable boundary layer (SBL) forms typically less than 0.5km above the ground (Garratt, 1992; Mehta et al., 2017). The mixing mechanisms are restricted within this layer during the night. Unfortunately, the
SBL cannot be detected with the lidar of Thessaloniki since most of the profiles start above 0.8km. Despite that, the particles that have been transported by the turbulence during the day take more time to settle, forming the so-called residual layer. As far as the aerosols are considered, this layer height bears many similarities to the daytime boundary layer height. We are particularly interested in this nighttime layer since the aerosol extinction coefficient profiles are available only after sunset (see section 3.2). Both for this reason and for reasons of simplification, in the next sections, we will use the terms "daytime PBL"
instead of daytime boundary layer and "nighttime PBL" instead of nighttime residual layer.

The upper boundary of the daytime and nighttime PBL was identified in approximately 99% of the cases. At this point it is necessary to mention that the PBL top is difficult to discern when large transported aerosol layers arrive and mix with local particles below 2km. In those cases, the PBL height can be either completely obscured or misidentified as the transported layer's upper boundary. Baars et al. (2008) present such an example. In one of their cases, an elevated dust layer complicated
the retrieval of the PBL height. Additionally, due to hardware restrictions of the lidar instruments, such as the system's overlap function (Wandinger and Ansmann, 2002), near ground values are typically not provided. As far as the system of Thessaloniki

is concerned, most of the profiles begin above 0.8 km. It is indeed quite rare to find profiles starting below 0.6 km. This, however, could also result in false identification of the PBL top when it is located close to the profile's starting height. This is expected to affect more the winter months, when the PBL is expected to be lower in Thessaloniki (Georgoulias et al., 2009). On the other hand, the winter measurements correspond to less than 10% of the profiles that were used for the PBL analysis.

### 3.3.2  Lofted layer height detection

An adaptation of the previous method (section 3.3.1) is applied on the lofted layers. Since this is a climatological study and the interest is not in the fine structure that individual profiles may exhibit, we decided to identify only the first three major lofted layers. For this reason, a dilation of 0.8 km has been used. Finally, the center of mass is calculated based on equation 2 in which COM is the center of mass, z is the altitude, F is the profile product that is used in order to obtain the geometrical

properties, while $z_b$ and $z_t$ are the layer's lower and upper boundaries respectively.

$$COM = \frac{\int_{z_b}^{z_t} z \cdot F(z) \cdot dz}{\int_{z_b}^{z_t} F(z) \cdot dz} \tag{2}$$

The first major layer was present in 48% of the profiles, while only 6% exhibited a second layer and much less a third layer. This is not surprising considering the large dilation value. A climatological analysis requires a sufficient number of data. This is the reason why we decided to exclude the second and third major layers from the analysis.

The results are presented in section 4.1. In section 3.4, the processes that took place in order to obtain additional optical products from the ones already available are discussed.

### 3.4  Optical properties

A subset of the full lidar dataset was utilized for the analysis of the aerosol optical properties, which includes the night-time aerosol extinction profiles at 355nm and the night-time aerosol backscatter profiles at 355nm (Raman inversion) and 532nm

(Klett inversion). We excluded the daytime backscatter profiles in order to be consistent with the extinction climatology, since the extinction profiles are only available during night-time. The lidar ratio (LR, equation 3) at 355nm and the backscatter related Angstrom exponent (BAE, equation 4) at the spectral range 355-532nm can be calculated from the initial products. The lidar ratio is produced solely from Raman profiles whereas the BAE 355-532nm is calculated both from Raman profiles, at 355nm, and from Klett profiles, at 532nm (see section 3.2). Both of these intensive properties are widely used because they are

independent of the aerosol concentration thus carrying information about the aerosol type and size. The respective formulas are provided in equations 3 and 4, where $\lambda$ is the wavelength, z is the height, a is the aerosol extinction coefficient, and b is the aerosol backscatter coefficient.

$$LR(\lambda, z) = \frac{a(\lambda, z)}{b(\lambda, z)} \tag{3}$$

$$BAE_{\lambda1-\lambda2}(z) = -\frac{ln(\frac{b(\lambda_2, z)}{b(\lambda_1, z)})}{ln(\frac{\lambda_2}{\lambda_1})} \tag{4}$$

Furthermore, some columnar products can be easily obtained from the profiles. The AOD and the mean columnar extinction at 355nm, as well as the integrated backscatter (INTB) and the mean columnar backscatter at 355nm and 532nm are calculated first. Then, the columnar lidar ratio at 355nm and the BAE at 355-532nm are produced from the mean extinction and backscatter values. Finally, the PBL top height (see section 3.3) is used in order to separate the boundary layer and the free troposphere. After this, the aforementioned columnar products can also be separately calculated inside these two atmospheric regions.

## 3.5  Data filtering and averaging

This study is focused on climatological cycles and trends. The occurrence of random rare events that greatly deviate from the standard behavior within a given time range can affect the representability of the monthly and seasonal averages. Consequently, a filter that excludes such extreme events is applied on all optical products. For each product population, the upper and lower quantiles are produced for each month. Values that exceed the upper and lower quantiles more than 1.5 times the interquantile range are excluded sequentially, one at a time, until there are no more outliers. Given, for instance, a normally distributed population, this filter would apply to the values that exceed approximately $\pm 2.7\,\sigma$, which corresponds to 0.7 % of the values. This applies to all the products described in sections 3.3 and 3.4. The backscatter and extinction profiles are filtered out based on their columnar versions, that is, the total AOD and the total integrated backscatter respectively. The filtering is applied to the daily averages of both the lidar and the sunphotometer datasets. Ultimately, the purpose of this process is to eliminate the effect of the extremes in the monthly and seasonal averaging.

In order to calculate the monthly and seasonal (DJF, MAM, JJA, SON) mean values from the filtered products, the daily means are calculated first. Then the monthly means for each year are calculated by averaging the daily means and the seasonal means are produced by averaging the monthly mean values. For the EARLINET dataset, every available night-time extinction profile at 355nm and every night-time backscatter profile at 355nm and 532nm (section 3.4) is used. For the AERONET dataset, however, a limit of at least 10 daily mean values per month and at least 2 out of 3 monthly values per season was set in order to ensure that the averages are representative enough. We have to clarify here that the aim of this study is not to make a point-by-point comparison of the two datasets but to compare two independently estimated climatologies. In all cases, a limit of at least 5 years of monthly or seasonal averages per annual value is set for the annual cycles and seasonal profiles. This limit is empirical. Its purpose is to increase the representativity of the annual cycle without loosing too many data points. Missing months or missing parts of the profile in figures 4 and 5, occur from this particular filter.

## 4  Results and discussion

The results of the climatological analysis of the optical and geometrical aerosol properties in Thessaloniki are presented in this section. The layer analysis of section 3.3 is displayed and discussed in section 4.1, while sections 4.2 and 4.3 include respectively information on the seasonal response of all the columnar and profile products under study respectively. Finally, the long-term trends of the two AOD databases are presented and discussed in section 4.4.

## 4.1 Layer analysis

In this section the distributions of the layer features are examined. Figure 3 on the left contains the results displayed in histograms for the daytime and nighttime PBL top height, while table 1 contains some metrics of the distributions. As it was mentioned in section 3.3.1, the daytime PBL corresponds to the available measurements between 10 UTC and 13 UTC, while the nighttime PBL corresponds to all the available measurements after sunset. The daytime boundary layer and night-time residual layer top is identified in 99% of the observations. The two distributions are similar with median values around 1.2 km. According to table 1, the median difference is quite small, less than 0.1 km. As mentioned in section 3.3.1, the SBL is undetectable with the lidar system since it is so close to the ground. There is a peak at 1.1 km which is more pronounced for the nighttime PBL distribution. Furthermore, the majority (more than 50%) of the cases exhibit PBL values between 0.9 and 1.8 km. It is important to mention that the PBL top could be misidentified when the real PBL top is located below 0.8 km because, as mentioned in section 3.3.1, the starting height of the profiles is typically above that height. This should mainly affect the winter measurements when the PBL top is expected to appear closer to the ground.

The results regarding the lofted layer are presented in figure 3 on the right. The upper and lower boundary as well as the center of mass distributions are displayed in histograms. All three of them are flatter than the PBL distribution, as the frequency never exceeds 15% in any height class. The maximum values appear at 1.7 km, 2.1 km, and 3.1 km and the median at 1.86 km, 2.49, and 3.14 for the base, center of mass, and top respectively. The layer thickness ranges between 0.69 km and 1.47 km for 50% of the cases. More information on the distributions is included in table 1. As stated in section 3.3.2, the lofted layer was present in 48% of the profiles. The seasonal analysis of the geometrical parameters displayed here is presented in section 4.2 along with the various retrievals from lidar data.

## 4.2 Seasonal cycles - Columnar Products

In this section the optical and geometrical properties are analyzed in order to detect seasonalities in their annual cycle. The extrapolated AOD at 355nm from the AERONET dataset is also included. The results of the columnar optical products and the geometrical products are displayed in monthly boxplots (figure 4) while the results of the profile optical products are exhibited in the form of seasonal average profiles (see section 4.3). The boxplots are constructed using the monthly average population. This is the reason why some outliers occur in figure 4 despite the application of the filtering process which has been applied to the initial and daily averages per month mentioned in section 3.5. The annual monthly averages are also included in figure 4 (dots).

### 4.2.1 Aerosol Optical Depth

The results from the AOD 355nm analysis are displayed in figure 4a and 3b. The AERONET dataset shows an annual cycle with the maximum annual mean values around 0.5 for July and August and the minimum values close to 0.25 in the winter months (figure 4a). A small secondary maximum appears at 0.4 in April. The EARLINET dataset shows a consistent annual cycle with the AERONET dataset. The Pearson correlation coefficient between the two annual cycles is 0.84, which is considered

high (e.g., Lolli et al., 2013). The annual mean lidar AOD values range from 0.2 in January to 0.65 in August. Higher lidar values are clearly observed during summer. Furthermore, the lidar values are more broadly distributed. They exhibiting always longer interquantile ranges, especially in April and the summer months. This probably occurs because the lidar sampling rate is much more sparse than the sunphotometer sampling rate. February and December are not included as the cloudy weather
conditions in the winter resulted in lidar data availability which does not fulfill the criteria mentioned in section 3.5. Apart form cloudy conditions, due to hardware limitations, it is not possible for the lidar system to operate during days with strong winds. This is not the case for the sunphotometer and, therefore, it could affect the results. For example, the AOD overestimation by approximately 0.1 of the lidar dataset during the summer months could be explained if days with strong winds in the summer are connected with lower aerosol load. Another probable explanation involves the uncertainties introduced due to the
system's overlap in combination with the use of night-time lidar measurements and daytime sunphotometer measurements. A systematic seasonal bias has been detected when isolating common sunphotometer and lidar cases and is discussed in section 4.5.1. It equals 0.13 during summer, corresponding to higher lidar AOD, and -0.15 during winter, corresponding to lower lidar AOD. Consequently, the summer and winter AOD differences observed in figure 4a could be attributed to such issues.

The AOD cycle in the PBL and in the FT is presented in figure 4b. The contribution from the free troposphere seems to be
comparable and even higher than the PBL contribution during April and the summer months. This is probably attributed to transported aerosols during summer and spring in the FT (see section 4.2.2.4). The other months, especially March, exhibit a lower FT contribution.

### 4.2.2   Lidar ratio and Backscatter related Angstrom

As far as the lidar ratio at 355nm and the BAE at 355-532nm is concerned, they exhibit more complicated patterns, ranging
from 45 to 70 sr and 1.0 to 2.0 respectively. The lidar ratio shows two peaks, one in the summer months and another one in November that probably extends to January (figure 4c). Unfortunately, this is not so clear since February and December are not included. The minimum values occur in the spring months and in the early autumn months. The BAE cycle, on the other hand is relatively stable, fluctuating between 1.1 and 1.5 for most months. The minimum values, that indicate larger particles, appear in May at 0.9, while the maximum values, that indicate smaller particles, appear in January at 1.9. Since both the lidar
ratio and the BAE depend mainly on the aerosol type and size and not on the concentration, their variability should be more sensitive to transported aerosol events. For example, the higher lidar ratio and BAE values observed in the summer months are indicative of mixing with biomass burning layers. On the other hand, smaller BAE values accompanied by smaller lidar ratio values could be the result of a stronger sea salt or dust component. The optical properties of the cases that are affected by layers of transported aerosol and their climatological behavior are presented and discussed in section 4.3.

### 4.2.3   Boundary Layer and First Lofted Layer

The PBL height and the lofted layer center of mass cycles are presented in 3e and 3f respectively. Looking at the PBL height, the maximum mean values, around 1.5 km, appear from May to September. The minimum values, close to 1.1 km occur in March and December. In general, the PBL seems to be higher in the warm months (May to September) and lower in the cold

months (November to March), as expected (Georgoulias et al., 2009), with the exception of January. This could be attributed to the difficulties that the lidar system faces below 800m that were discussed in sections 2.2 and 3.3.1. Additionally, it was mentioned above that the lidar system usually operates under cloud free conditions. In winter, this could result in a sampling that favors the presence of high pressure systems and consequently higher PBL top height values. The missing point in February just makes it more difficult to draw any firm conclusions on this. The lofted layer is higher from February to September with two peaks at May and August, probably due to dust and biomass burning layers that arrive in the FT. The lowest values appear in January and December.

## 4.3 Seasonal Cycles - Profile products

In this section, the seasonal profiles of the extinction coefficient at 355nm, the lidar ratio at 355nm, and the BAE at 355-532nm are discussed. The results are presented in figure 5 and in tables 2, 3 and 4. The seasonality of each product is also analyzed in the boundary layer and the free troposphere per mixture type. These results are presented in tables. Four categories are included. The category "all" corresponds to the whole dataset for the optical properties (see section 3.4). The categories "dust mixtures" and "biomass mixtures" correspond to the cases that contain at least one transported Saharan dust or biomass burning layer respectively. The category "continental" or "cont" contains all the rest of the cases. This can include mixtures of soil dust, urban, agricultural or maritime aerosol. The characterization of the dust and biomass burning measurements is already available in the EARLINET database, since it is performed manually per station before the measurements are uploaded. The process includes a back-trajectory analysis from the Hybrid Single Particle Lagrangian Integrated Trajectory Model HYSPLIT per layer. The biomass burning activity along the trajectory path is examined using fire pixel data from the MODIS Terra and Aqua Global Monthly Fire Location Product (MCD14ML). The presence of dust particles for trajectories passing over the Sahara desert is cross-checked using model simulations from the Dust Regional Atmospheric Model (BSC-DREAM8b). Even one transported layer in a profile is enough to flag the measurement. Consequently, the "dust mixtures" and "biomass mixtures" profiles are seldom pure. They are expected to be mixed with continental aerosol, especially near the ground where the local particles are more dominant. Another type of special event that is available in the database is the volcanic category. For Thessaloniki, this mainly includes some cases of transported volcanic ash during April and May 2010 when the Eyjafjallajökull volcano erupted in Iceland (Pappalardo et al., 2013). These volcanic cases have not been included in a separate mixture category since this type of aerosol mixture is too rare.

### 4.3.1 All cases

The aerosol extinction coefficient at 355nm is maximum in summer and minimum in winter (figure 5.i.a) for the category "all". The AOD at 355nm reaches 0.30 both in the PBL and in the FT during summer (table 2). In winter, those values decrease to 0.14 and 0.08 respectively. The lidar ratio ranges mostly between 49 to 61 sr (table 3) for this category. The minimum values appear during spring and the maximum during summer. The BAE, on the other hand, ranges mostly from 1.0 to 1.7 and the biggest particles tend to appear during autumn and spring in the PBL, while the smallest ones during winter in both atmospheric regions (table 4).

### 4.3.2 Continental

When the dust and biomass burning episodes are excluded ("cont" category), the extinction profile of spring decreases down to the winter levels (figure 5.ii.a). The spring AOD drops from 0.20 and 0.16 to 0.12 and 0.11 in the PBL and in the FT respectively (table 3). The other seasons are not affected as much. The lidar ratio ranges from 47 to 61 sr (table 4). Giannakaki
et al. (2010) report an annual mean value of $56 \pm 23$ sr for the continental polluted particles in Thessaloniki during the period 2001-2007. This comparison, however, is not completely straightforward for the continental particles, since in their study they divide them in three subcategories (local, continental polluted, and continental west/northwest) based on the wind direction. This is not performed here. The minimum values at 46 sr appear in spring. This could be attributed to mixing with maritime aerosol. It is within the range that Burton et al. (2012) report for polluted maritime particles. The other seasons are within the
range that Burton et al. (2012) report for urban particles. Autumn and winter exhibit the highest variability. The BAE values range between 1.7 and 1.9 for all seasons except autumn (table 5). The minimum values are observed at 0.9 during autumn in the PBL. According to Heese et al. (2017) lower Angstrom values are more typical of pollution mixtures rather than of pure pollution. Giannakaki et al. (2010) report an annual mean value of $1.4 \pm 1.0$ for the continental polluted aerosol.

### 4.3.3 Dust mixtures

As far as the "dust mixtures" group is concerned, the maximum values in the extinction profiles at 355nm appear in summer above 1.5 km. High values also appear in autumn in the near range (figure 5.iii.a). The AOD values range from 0.17 to 0.31 (table 2). Unfortunately, the winter extinction profile is missing, since the dust cases are rare during this season in Thessaloniki. The autumn data availability is also marginal. The lidar ratio at 355nm ranges from 47 to 61 sr (table 3). Giannakaki et al. (2010) report an annual value of $52 \pm 18$ sr. The minimum values occur during spring, and during autumn in the PBL, ranging
between from 45 to 48 sr. These values are typical of dust and marine mixtures (Groß et al., 2015; Mona et al., 2006). The summer values at 60 and 61 sr in the PBL and in the FT respectively seem closer to the expected values for transported dust (Groß et al., 2015). It is possible that the wind circulation is responsible for this behavior. Due to a high pressure system over the Balkans that occurs typically from May to September (Tyrlis and Lelieveld, 2013), it is more difficult for the dust layers to be transported directly from Northwestern Africa to Thessaloniki through southwest winds that pass over the Mediterranean.
Consequently, the dust particles are forced to travel a longer path, through central Europe in order to reach Thessaloniki (Israelevich et al., 2012). This behavior could result in the different mean lidar ratios between summer and the other two seasons. The BAE ranges mostly between 0.9 and 1.2 (table 4), values that are typical of dust mixture (Papayannis et al., 2009; Baars et al., 2016). Giannakaki et al. (2010) report an annual BAE value of $1.5 \pm 1.0$ sr for this category. A summer BAE of 1.6 in the PBL versus 1.2 in the FT indicates that, in the PBL, the particles are either quite mixed or absent. In the FT the dust
component can still be considered dominant, since the BAE is shifted towards values closer to the transported dust Angstrom of $0.5 \pm 0.5$ reported within EARLINET (Müller et al., 2007). Indeed, Marinou et al. (2017) show that the dust component during the transportation episodes over Greece in summer (JAS) is more dominant above 2km during summer which is consistent with our findings.

#### 4.3.4 Biomass burning mixtures

The main source of biomass burning aerosol for Thessaloniki is agricultural fires in the Balkans, Belarus and European Russia that typically begin after March and end in October (McCarty et al., 2017; Amiridis et al., 2009). These mixtures exhibit vertical distributions with maximum values during summer. Below 1km, the spring and autumn profiles are quite similar. The AOD 355nm generally ranges from 0.18 to 0.24 with the exception of summer in the FT where the largest AOD value of table 3 occurs at 0.39. It is possible that the strong biomass burning events tend to occur during summer and the smoke aerosols are usually transported at higher altitudes. Winter is entirely missing here as well, since the weather conditions are unfavorable for fires. The lidar ratio ranges from 51 to 73 sr. The highest values, above 70 sr appear during summer while the minimum lidar ratio is observed in the PBL during spring. It is close to the respective continental lidar ratio and also within the range that Heese et al. (2017) report for pollution particles. Consequently, it is quite possible that the biomass layers affect less, if not at all, the boundary layer during spring. In all other cases, the lidar ratio is similar, ranging from 59 to 61 sr. Differences with the summer levels could be attributed to different aerosol transportation paths and thus either more mixing with continental particles or different aging of smoke (e.g., Amiridis et al., 2009; Nicolae et al., 2013; Papayannis et al., 2014). The BAE values are available only for summer and autumn, ranging from 1.3 to 1.4. Giannakaki et al. (2010) report an annual mean lidar ratio of $69 \pm 17$ sr and a mean BAE of $1.7 \pm 0.7$ for this category which seems consistent with our results.

#### 4.4 Long-term changes

The AOD at 355nm is selected for the timeseries analysis, since it is the product with the longest data span for both the EARLINET and the AERONET datasets. The two timeseries of seasonal averages are shown in figure 6a. The lidar AOD values cover a larger range and show higher variability than the sunphotometer values. This is expected given the much lower data availability in this dataset. We intend to compare the two timeseries in terms of trends and not point by point. The linear fit slope values seem consistent for the two timeseries. The EARLINET dataset results in a decrease of the AOD by 0.0109 per year while the sunphotometer dataset in a decrease of 0.0075 per year. This translates to a decrease per decade of 29.0% versus 20.7% respectively compared to the AOD levels in 2003. In order to calculate the long-term trend during the period 2003-2017 the seasonality must be removed from the timeseries. This is performed by subtracting the respective seasonal annual cycle from each year for both datasets. The resulting values are the seasonal AOD anomalies. These timeseries are presented in figure 6b. The least square fit slope here represents the dataset trend. The new values are -0.0088 (23.2%) and -0.0081 (22.3%) in the period 2003-2017 for the EARLINET and the AERONET datasets respectively. (Fountoulakis et al., 2016) report a negative AOD 320nm trend of -0.009 per year for Thessaloniki during the period 1994-2014, a result that seems consistent with our findings. We have applied a Mann-Kendal non-parametric test in order to ensure the existence of these trends (Hirsch et al., 1982; Gilbert, 1987). Both of them are statistically significant at the 95% confidence interval. We further investigate this decreasing trend by looking at the AOD timeseries in the PBL and in the FT that are available for the EARLINET dataset. The two products are directly compared in figure 6c and their seasonal anomalies are presented in figure 6d. It appears that the free tropospheric AOD slightly increases by 0.0016 per year, however this trend is not statistically significant at the 95 %

confidence interval. The PBL AOD, on the other hand, shows a decreasing statistically significant trend of -0.0104 per year. Consequently, the decrease of the total AOD seems to occur mainly in the lower atmospheric layers, inside the PBL. This could be attributed to a reduction of the aerosol load coming from local sources. A change in the aerosol type, such as a shift to less absorptive particles in the PBL could also be responsible for this behavior (Fountoulakis et al., 2016). Further research on the aerosol microphysical properties could contribute to gain insight into this matter.

## 4.5 Factors affecting the compatibility of the two climatologies

In this section, we present some diagnostic tests that have been performed in order to ensure that the two climatologies can be safely compared despite the different sampling and the non-simultaneous acquisition of measurements. In section 4.5.1, periodical systematic biases that could affect the annual cycles are discussed. Non-periodical biases that could interfere with the long-term trends are addressed in section 4.5.2. Finally, section 4.5.3 includes an analysis of issues that arise due to the different sampling rate between the lidar and the sunphotometer.

## 4.6 Seasonal systematic biases

Since the sunphotometer measurements are performed during the day and the lidar Raman measurements during the night, a systematic bias could be introduced due to daily cycles of emission and meteorology. Additionally, the lidar profiles seldom extend below 600m. This could also contribute to a systematic bias. This bias is expected to produce an offset and/or seasonal discrepancies between the two datasets. In order to investigate the aforementioned issues the common daily averages between the two datasets are isolated in order to ensure that only the overlap issues and the day/night discrepancies would contribute to the bias. We have computed the AOD at 355nm biases by subtracting the sunphotometer daily mean AOD from the lidar daily mean AOD per case. The seasonal biases and the total bias are calculated with a methodology similar to the one applied to the lidar measurements (see section 3.5). The daily means are calculated first. Then the monthly means for each year are calculated by averaging the daily means and the seasonal means are produced by averaging the monthly mean values. Spring and autumn biases are close to zero with values at 0.03 and -0.01 respectively. The winter seasonal bias is -0.15 while the summer bias is 0.13. The total bias is close to zero, at -0.003. Consequently, there is a minor offset towards slightly lower lidar AOD values between the two annual cycles and a systematic estimation of higher lidar AOD values in summer and lower lidar AOD values in winter. This behavior is already visible in the monthly annual cycles (figure 4), especially for summer.

## 4.7 Non-periodical systematic biases

As far as the long term trend analysis is concerned, even if the sunphotometer and the lidar AOD exhibit different seasonal patterns, the trend values should not be much affected since the seasonality has been removed from each timeseries individually (see section 4.4). Furthermore, an artificial trend could also be introduced to the lidar timeseries if the bias is non-periodically time-dependent. Changes in the system's full overlap height (see section 2.2) within the timeseries could produce such an effect. We examine such effects by calculating the trend of the seasonal bias after removing the bias seasonality. The deseasonalized

bias exhibits a negative trend of 0.0022 per year, however, it is not significant. As a result, the long term trend of the lidar AOD is not significantly affected by systematic biases.

## 4.8 Sampling

Another issue that needs to be addressed is that the sparse EARLINET sampling could result to averages that are not representative and comparable to the AERONET ones. This would significantly affect the annual cycle and trends. We limited the AERONET dataset to only Monday and Thursday measurements to be compatible with the EARLINET schedule of night-time measurements. The resulting significant trend is -0.0090 per year, very close to -0.0085 that occurs when using the whole dataset (figure 7). The annual cycle seems stable with absolute differences smaller than 0.08 for every monthly average. To be on the safe side, we obtained the sunphotometer trend using only the daily means where both a sunphotometer and a lidar measurement were available. The resulting significant trend is -0.0089 per year (figure 7), still close to -0.0085 that occurs when using the whole dataset. Consequently, the lidar averages should be statistically meaningful and the uncertainty in the EARLINET trend should be less than ±0.0005 due to the limited sampling. Probably the length of the timeseries (14 years) compensates the sparse sampling rate. In the future, we plan to further analyze how the sampling and the timeseries length affect the climatological products produced from the columnar aerosol optical properties.

## 5 Conclusions

The analysis resulted in consistent, statistically significant, and decreasing seasonal AOD 355nm trends of -23.2% and -22.3% per decade in the period 2003-2017 over Thessaloniki for the EARLINET and the AERONET datasets respectively. This implies that the EARLINET schedule of data acquisition can be quite effective in producing data that can be applied to climatological studies. Furthermore, the decreasing trend observed is mainly attributed to changes in the aerosol properties inside the boundary layer. The free tropospheric AOD, on the other hand, does not change much in the period under study and this change is also not statistically significant. This behavior could be attributed to either changes in the local emissions or in the aerosol type inside the PBL. Further investigation is required on this, however. Concerning the seasonal profiles of the period 2003-2017, the highest values of the extinction at 355nm appear during summer while the lowest ones appear during winter. The mean lidar ratio ranges between 47 sr and 61 sr for the continental particles. Mixing with Saharan dust and biomass burning aerosol is rare during winter. The dust component is more dominant in the FT than in the PBL during summer. This behavior is supported by other studies. In spring and autumn, the lidar ratio is approximately 47 sr which is more typical of dust and marine mixtures. Concerning the biomass burning cases, the transported layers tend to arrive in the FT during spring and summer. Lidar ratio values close to 60 sr are observed during autumn and during spring in the free troposphere. It increases to approximately 72 sr in summer, which could be the result of different smoke aging caused by different wind circulation paths. Such seasonal profiles of the most dominant aerosol types can be quite useful for applications that require a priori aerosol profiles, for example, they can be utilized in models that require an aerosol climatology as input, in the development of algorithms for satellite products, and in passive remote sensing techniques that require the information of the aerosol vertical

distribution. Future studies that focus on the climatological circulation patterns of the air masses that arrive in Thessaloniki will reveal more information on the seasonal variations of the aerosol properties that are observed and discussed here.

*Data availability.*

The lidar data used in this study are available upon registration at http://data.earlinet.org. The AERONET sunphotometer
data for Thessaloniki are publicly available at https://aeronet.gsfc.nasa.gov/.

*Competing interests.*

The authors declare that they have no conflict of interest.

*Acknowledgements.* This work has been conducted in the framework of EARLINET (EVR1 CT1999-40003), EARLINET-ASOS (RICA-025991) ACTRIS and ACTRIS-2 funded by the European Commission. The research leading to these results has received funding from the
European Union's Horizon 2020 research and innovation programme under grant agreement No 654109 and previously from the European Union Seventh Framework Programme (FP7/2007-2013) under grant agreement No 262254. Elina Giannakaki acknowledges the support of the Academy of Finland (project no. 270108). Kalliopi A. Voudouri acknowledges the support of the General Secretariat for Research and Technology (GSRT) and the Hellenic Foundation for Research and Innovation (HFRI). Konstantinos Fragkos would like to acknowledge the support from European Union's Horizon 2020 Research and Innovation Programme, under Grant Agreement no 692014 - ECARS. This
research has been co-financed, via a programme of State Scholarships Foundation (IKY), by the European Union (European Social Fund - ESF) and Greek national funds through the action entitled "Scholarships programme for postgraduates studies - 2nd Study Cycle" in the framework of the Operational Programme "Human Resources Development Program, Education and Lifelong Learning" of the National Strategic Reference Framework (NSRF) 2014 – 2020.

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

**Table 1.** Metrics of the aerosol geometrical properties.

|  | Median | Upper Quantile (75%) | Lower Quantile (25%) | Interquantile Range | Upper Wisker | Lower Wisker |
|---|---|---|---|---|---|---|
| Day PBL | 1.22 | 1.62 | 0.98 | 0.64 | 2.51 | 0.74 |
| Night PBL | 1.25 | 1.72 | 0.96 | 0.75 | 2.78 | 0.71 |
| Layer Base | 1.86 | 2.55 | 1.61 | 0.94 | 3.92 | 0.98 |
| Center of Mass | 2.49 | 2.99 | 2.03 | 0.96 | 4.20 | 1.35 |
| Layer Top | 3.14 | 3.74 | 2.49 | 1.25 | 5.03 | 1.79 |
| Thickness | 0.91 | 1.47 | 0.69 | 0.78 | 2.55 | 0.33 |

**Table 2.** Mean values and variability of the aerosol optical depth at 3555nm in the boundary layer and in the free troposphere. This seasonal values are produced from the respective monthly mean averages.

| Season | Type | All | Cont. | Dust Mix. | Biom. Mix. |
|--------|------|-----|-------|-----------|------------|
| | | **Aerosol Optical Depth at 355nm** | | | |
| Winter | PBL | $0.14 \pm 0.09$ | $0.14 \pm 0.09$ | - | - |
| | FT | $0.08 \pm 0.02$ | $0.08 \pm 0.02$ | - | - |
| Spring | PBL | $0.20 \pm 0.09$ | $0.12 \pm 0.05$ | $0.23 \pm 0.08$ | $0.20 \pm 0.10$ |
| | FT | $0.16 \pm 0.07$ | $0.11 \pm 0.05$ | $0.17 \pm 0.08$ | $0.18 \pm 0.11$ |
| Summer | PBL | $0.30 \pm 0.16$ | $0.28 \pm 0.23$ | $0.31 \pm 0.15$ | $0.24 \pm 0.07$ |
| | FT | $0.30 \pm 0.07$ | $0.27 \pm 0.08$ | $0.29 \pm 0.11$ | $0.39 \pm 0.09$ |
| Autumn | PBL | $0.18 \pm 0.10$ | $0.16 \pm 0.09$ | $0.31 \pm 0.17$ | $0.23 \pm 0.09$ |
| | FT | $0.15 \pm 0.05$ | $0.12 \pm 0.04$ | $0.28 \pm 0.13$ | $0.21 \pm 0.12$ |

**Table 3.** Mean columnar values and variability of the lidar ratio at 355nm in the boundary layer and in the free troposphere. This seasonal values are produced from the respective monthly mean averages.

| | | Lidar Ratio at 355nm [sr] | | | |
|---|---|---|---|---|---|
| Season | Type | All | Cont. | Dust Mix. | Biom. Mix. |
| Winter | PBL | $55 \pm 19$ | $56 \pm 19$ | - | - |
| | FT | $57 \pm 21$ | $57 \pm 21$ | - | - |
| Spring | PBL | $49 \pm 11$ | $47 \pm 14$ | $47 \pm 13$ | $51 \pm 12$ |
| | FT | $51 \pm 12$ | $46 \pm 11$ | $47 \pm 13$ | $61 \pm 10$ |
| Summer | PBL | $61 \pm 9$ | $60 \pm 15$ | $60 \pm 14$ | $73 \pm 10$ |
| | FT | $61 \pm 9$ | $61 \pm 15$ | $61 \pm 21$ | $71 \pm 7$ |
| Autumn | PBL | $53 \pm 17$ | $51 \pm 21$ | $45 \pm 13$ | $59 \pm 4$ |
| | FT | $57 \pm 16$ | $58 \pm 26$ | $48 \pm 15$ | $61 \pm 5$ |

**Table 4.** Mean columnar values and variability of the backscatter related Angstrom exponent 355-532nm in the boundary layer and in the free troposphere. This seasonal values are produced from the respective monthly mean averages.

| | | Backscatter related Ang. Exponent 355-532nm | | | |
|---|---|---|---|---|---|
| Season | Type | All | Cont. | Dust Mix. | Biom. Mix. |
| Winter | PBL | $1.6 \pm 0.6$ | $1.7 \pm 0.6$ | - | - |
| | FT | $1.7 \pm 0.3$ | $1.8 \pm 0.6$ | - | - |
| Spring | PBL | $1.2 \pm 0.8$ | $1.7 \pm 0.4$ | $1.0 \pm 0.9$ | - |
| | FT | $1.4 \pm 0.5$ | $1.7 \pm 0.7$ | $0.9 \pm 0.6$ | - |
| Summer | PBL | $1.5 \pm 0.6$ | $1.8 \pm 0.6$ | $1.6 \pm 0.5$ | $1.3 \pm 0.8$ |
| | FT | $1.3 \pm 0.4$ | $1.9 \pm 0.5$ | $1.2 \pm 0.6$ | $1.4 \pm 0.3$ |
| Autumn | PBL | $1.0 \pm 1.0$ | $0.9 \pm 1.1$ | $1.0 \pm 0.7$ | $1.4 \pm 0.4$ |
| | FT | $1.3 \pm 0.5$ | $1.1 \pm 0.8$ | $1.5 \pm 0.6$ | $1.4 \pm 0.2$ |

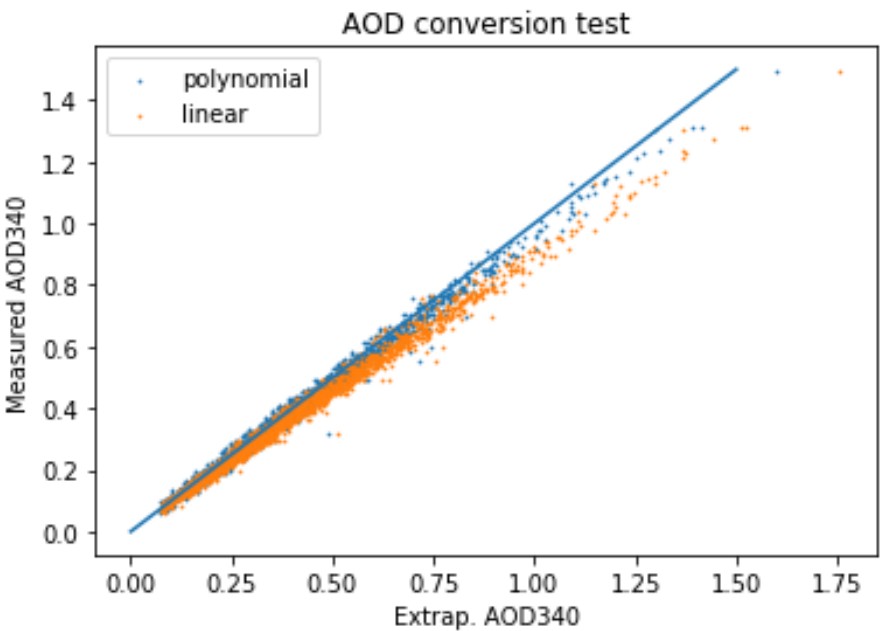

**Figure 1.** Scatter-plot of the measured sunphotometer AOD at 340nm against the extrapolated AOD at 340nm. Two methods of extrapolation are presented. The 'linear' approach assumes a linear behavior of the logarithm of the AOD in the spectral region 340-1020nm, while the polynomial approach assumes a 2nd order polynomial behavior. The unity line is also included.

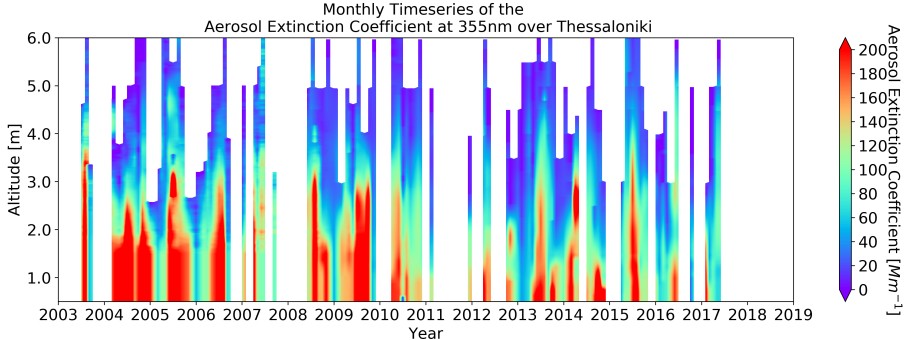

**Figure 2.** Time-height cross section of the monthly mean aerosol extinction coefficient at 355nm in the period 2003-2017.

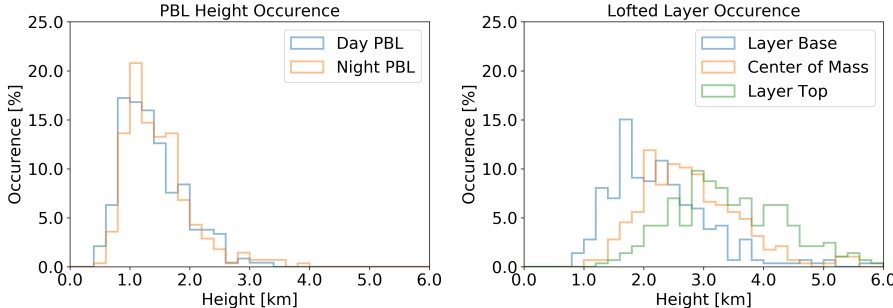

**Figure 3.** Histograms of the Daytime and Nighttime PBL top (left) and the first lofted layer base, center of mass and top height distributions (right). The height classes range is set to 200 m.

**Annual monthly boxplots of some of the optical and geometrical aerosol properties in Thessaloniki.**

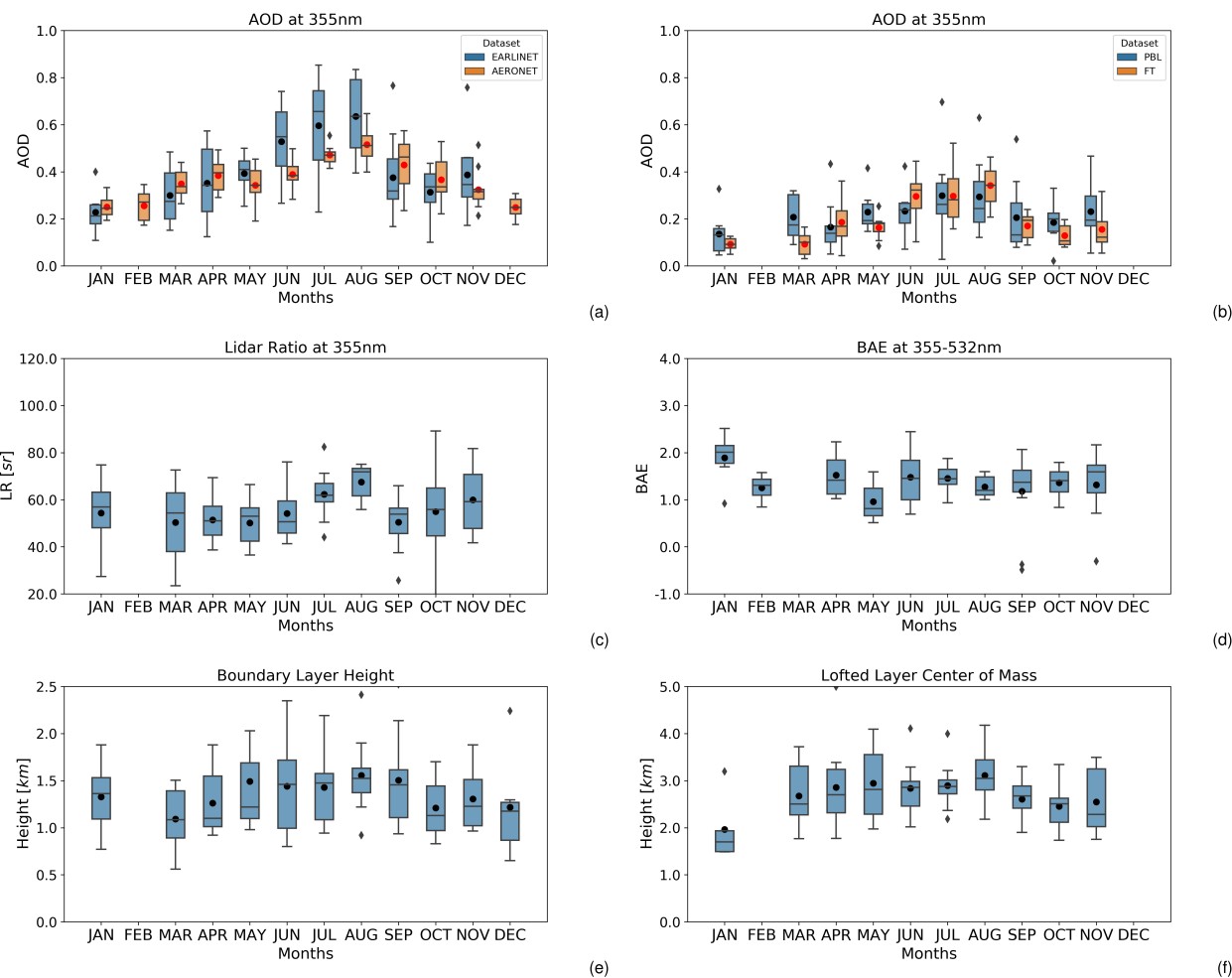

**Figure 4.** The annual cycle of the monthly mean columnar products. The AOD at 355nm in the whole column (a) but also in the PBL and the FT (b), the mean lidar ratio at 355nm (c), the mean BAE at 532-532nm (d), the mean PBL height (e) and the mean lofted layer center of mass (f) are included in this figure. The AERONET mean AOD at 355nm is also displayed in (a). In our analysis, the boxplot whiskers correspond to the most distant value encountered within 1.5 times the interquantile range above the upper and lower quantiles.

## Seasonal mean optical properties profiles for Thessaloniki

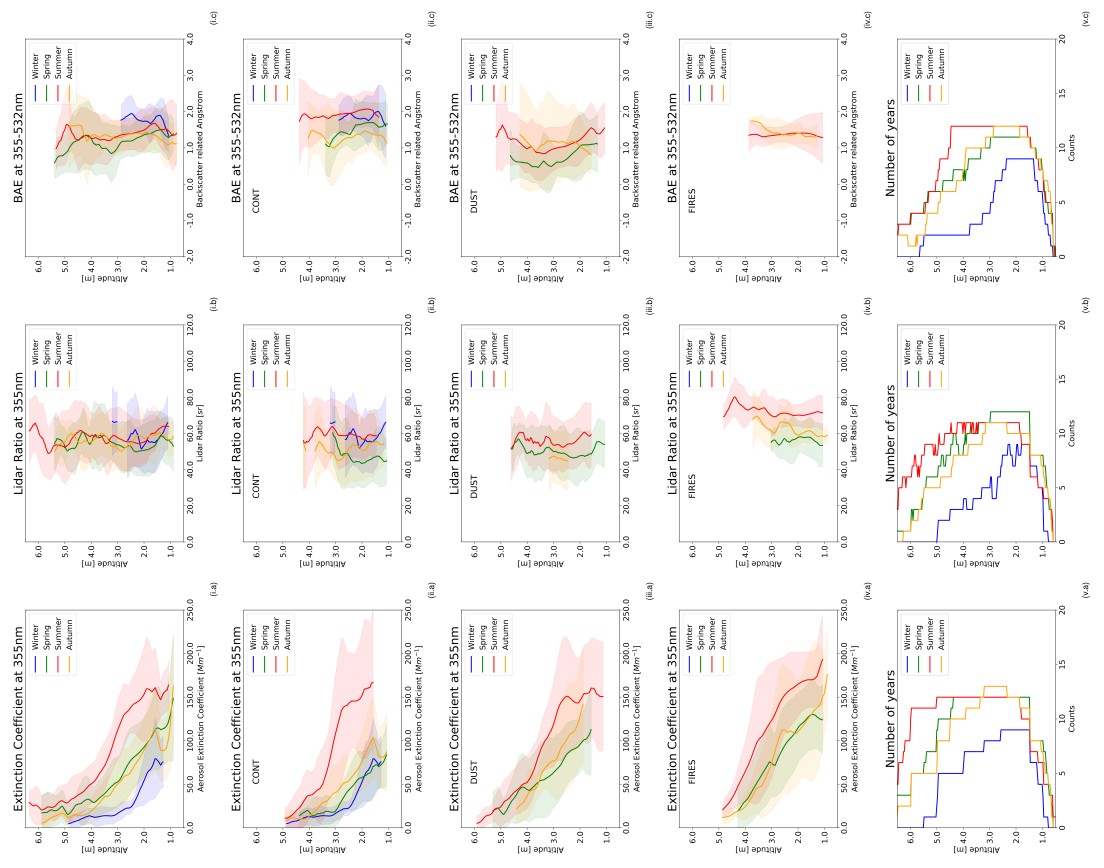

**Figure 5.** Seasonal profiles of the main aerosol optical properties under study. Rows (i), (ii), (iii), and (iv) correspond to the measurement categories "all", "continental", "dust mixtures", and "biomass mixtures" (see section 4.2.2) respectively while row (v) corresponds to the number of measurements profiles of the category "all". The profiles of the extinction coefficient at 355nm, the lidar ratio at 355nm and the BAE at 355-532nm are presented in columns (a), (b), and (c) respectively.

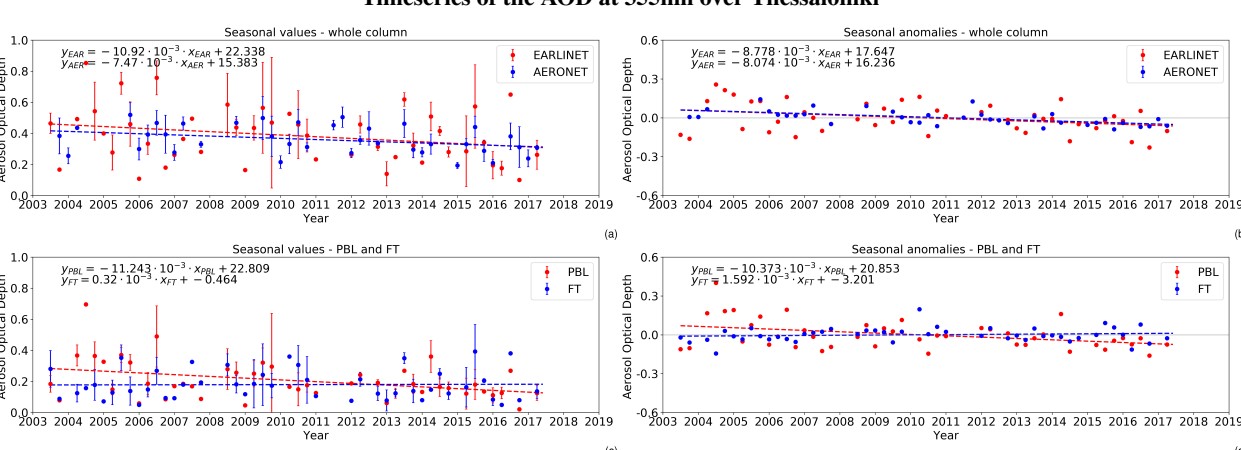

**Figure 6.** Timeseries of the seasonal mean AOD values at 355nm (a) and of the respective seasonal anomalies (b) that are produced after removing the seasonality for the whole column. The AERONET dataset is displayed along the EARLINET dataset for (a) and (b). Similar timeseries from the EARLINET dataset AOD in the PBL and in the FT are presented in (c) and (d) for the mean values and the anomalies respectively. The linear fit line is also included in the figures. For (b) and (d) it represents the AOD 355nm trend in the period 2003-2017.

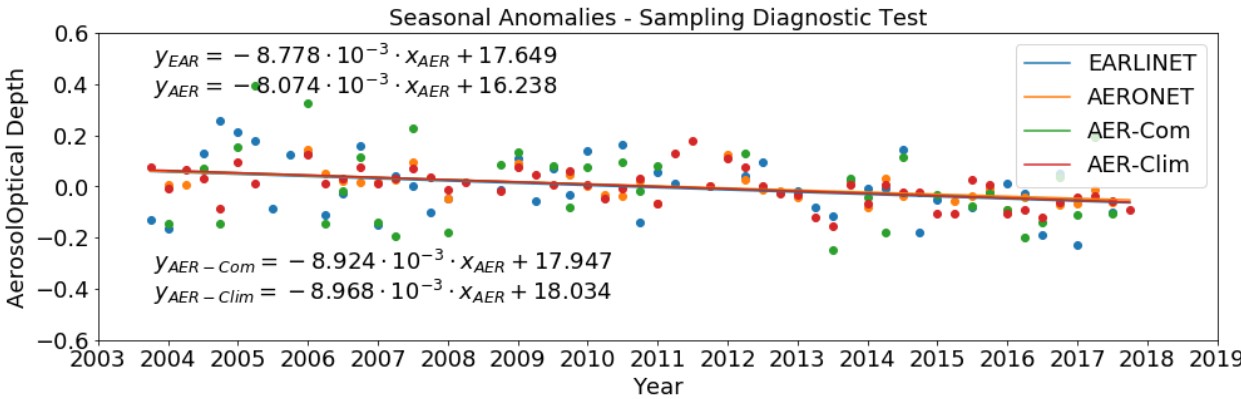

**Figure 7.** Timeseries of the seasonal AOD anomalies at 355nm. The original EARLINET timeseries is marked with blue while the original AERONET timeseries with orange. Two different sampling tests are performed on the AERONET dataset. The "AER-Clim" timeseries contains only Monday and Thursday measurements and it is marked with red while the "AER-Com" timeseries contains only common lidar and sunphotometer cases and it is marked with green.