# Peer review of "Are EARLINET and AERONET climatologies consistent? The case of Thessaloniki, Greece"

_Atmospheric Chemistry and Physics, 2017_

## Referee Comment (RC1) · Anonymous Referee #2 · 13 Mar 2018

In this study the authors present results from the climatological behavior of the aerosol optical properties over Thessaloniki during the years 2003-2017. Two independent datasets, representing two individual networks, the EARLINET and the AERONET, were applied to investigate the consistency and the statistical significance between both networks using geometrical and optical properties of aerosols. The analysis show a decreasing on AOD at 355 nm trends of -21.0% and -16.6% per decade for the EARLINET and the AERONET, respectively. Also, results show the dominance of dust and biomass burning on the free troposphere during summer. Different from other studies that considered only short time periods such as four or six years, and only one single kind of instruments (Lidar Raman), this study presented very important results of climatological studies of 14 years using two well establish networks. Overall,

the manuscript is well well-organized and clearly presented. I'd like to suggest the acceptance of this manuscript after some revisions.

Section 2.1 The Lidar setup – page 3 – lines 16 to 19.

The authors use the Lidar data set between 2003 to 2017 and states, "since a long timeseries of data was necessary, only the extinction 355nm and the backscatter 355nm and 532nm products were included in the analysis. The dataset included in this study covers the period 2003-2017 in order to be chronologically consistent with the sunphotometer dataset."

The Lidar dataset used is from 2003 to 2017. It is well known that EARLINET has a weel stablish standard pattern of quality assurance tests such as dark current, bin-shift, zero bin, trigger delay corrections, Telecover tests, Rayleigh fit, etc. Since when these tests are applied to Thessaloniki EARLINET station? Since 2003? What is the influence of these tests on the results of your comparisons? What type of errors or uncertainties the lack of these tests for the early dataset can take into account?

Section 2.2 The sunphotometer - page 3 – lines 25 to 26

"The level 1.5 aerosol optical depth values (AOD) at 440nm and the angstrom exponent 440-670 during the period 2003-2017 were used in this study. The AOD at 440nm is preferred for the comparison with the lidar UV products in order to take advantage of the longer time series since the 340nm and 380nm channels were added in 2005."

Why not to use Level 2.0 data? What would be the differences on the trend results using the level 2.0 since it is quality assured; the final post-deployment calibration values are applied to the data set, and the aerosol optical depth data are inspected for possible cloud contaminated outliers.

For AERONET level 1.5 data, when Angstrom parameter computed using all available channels between 440 and 870 nm is greater than -0.1 the point is considered cloud and pointing error free. Is the Level 1.5 AERONET data used for this study filtered

using this assumption?

Section 2.2 The sunphotometer - page 3 – lines 26 to 28

"The AOD at 440nm is preferred for the comparison with the lidar UV products in order to take advantage of the longer time series since the 340nm and 380nm channels were added in 2005."

You add 2 year more on your climatology (2003 and 2004, since the 340nm and 380nm channels were added in 2005). How is the difference in your result considering these 2 years more?

Subsection 4.2.1 - Aerosol Optical Depth – page 9 – 20 to 23

"The AOD cycle in the PBL and in the FT is presented in figure 3b. The contribution from the free troposphere seems to be comparable and even higher than the PBL contribution during April and the summer months. This is probably attributed to transported biomass burning aerosol during summer and spring in the FT (see section 4.2.2.4) The other months, especially March, exhibit a lower FT contribution."

It is possible to obtain some result or correlation of the biomass burning aerosol transported on the free troposphere using only AERONET AOD values? Or considering the annual cycle of the monthly mean columnar products of AOD at 355nm in the whole column presented on figure 3 (a), is possible that AERONET is missing any aerosol layer on the free troposphere? How could it affect the results of the decreasing trends?

Subsection 4.2.2 Integrated Backscatter – page 9 – lines 25 to 27

"Another columnar optical product, the integrated backscatter (INTB) at 355nm and at 532nm, is presented in figure 3c and 3d. The AERONET equivalent is calculated by dividing the AOD at 355nm and at 532nm with a constant lidar ratio of 50 sr and it is also included in the figures."

What kind of error uncertainties and/or bias the authors could expect using the fixed

Lidar ratio of 50 sr to calculate in INTB for the AERONET data? Since the the lidar ratio at 355 nm ranging from 45 to 70 sr according to statement on lines 2 and 3 of page 10, why not to use a mean fixed lidar ratio of 57 or 58 sr to calculate the AERONET integrated backscatter?

One thing that is not clear on the manuscript is the consideration about the column AOD comparison between AERONET data, that performs measurements during daytime, and the AOD from Raman Lidar measured during the nighttime. What kind of correction or assumption the authors take into account for these cases?

---

## Referee Comment (RC2) · Anonymous Referee #1 · 21 Mar 2018

The subject of the manuscript is relevant to the journal, as different end-users need vertically-resolved aerosol profiles obtained from climatological observations, instead of using models (i.e. ARMA model). Nevertheless the paper shows conceptual errors that introduce serious issues that makes it not suitable for publication. Please refer to comment section for details.

Major Flaws:

1) The two climatologies cannot be compared and no conclusion can be drawn. AERONET is a daytime measurement, while lidar observations are taken and averaged independently, both during daytime and nighttime. For sure, being different at night and day both the atmospheric conditions and aerosol emission sources (e.g. traffic and or household heating), a non-negligible bias is introduced in the analysis and

consequently it is not possible to establish whether the correlation is good or not. 2) In the text it is clearly specified that a mean value for the lidar optical thickness is obtained by averaging measurements based on the elastic scattering technique with those obtained with the Raman scattering technique. The use of different techniques introduces a further bias. 3) The most important contribution to the aerosol backscattering and extinction coefficients is coming from the first hundred meters that are heavily affected by overlap function. Only marginally In section 3.2, line 14 pag. 6 overlap problems are described . As nighttime and daytime profiles are averaged together, an additional source of bias is introduced: what about the profiles for which the aerosol load is confined below 500m? It looks like those profiles cannot be compared at all with AERONET retrievals as in fact only a portion of those aerosol layer is detected. 4) Even for daytime profiles, in the manuscript it is not even specified if an overlap correction is performed (i.e. shooting the lidar horizontally) and what is the extent of the lidar blind region and what the authors did to overcome this problem. 5) It seems that the comparison has been performed based on data from the EARLINET database. In spite of points 1) and 3) above, the comparison has been done considering on average 52 days per year (corresponding to Monday morning schedule). On 52 these days, how many of them are cloud free? Are then the averages statistically meaningful? The paper is missing such analysis.

Specific Comments:

Line 2 Pag. 1 Measurements are not deployed, instruments are.

Line 3 Pag. 1 Please read: "These two instruments are members of two different networks..."

Line 4 Pag. 1 Please read:" The instruments are operated under a different time schedule."

Line 12 Extinction is not defined. It is clear in the lidar community but for general audience it should be given a broader definition, as the vertical-resolved extinction

coefficient

Line 16 Pag. 1 Please read:"a priori climatological profiles" Line 16 Pag. 1 Please read: " they can be used by modelers community"

The English in the abstract was improved. This should be extended on the whole manuscript. Often sentences are too long and convoluted.

Line 1 Pag. 2 atmospheric particles don't show variability, but concentrations or load yes.

Line 4 Pag.2 atmospheric conditions is more appropriate than wind circulation, other phenomena as convection are important.

Line 14 Pag.2: "The in situ technique. . ." please rephrase as the sentence is not clear.

Line 17 Pag. 2. References are not at all exhaustive. This comment is valid through all the manuscript.

Line 31 Pag. 2 Raman indicates a person last name, then should be caps lock everywhere in the text.

Line 33 Pag. 2 As written before, it is missing an analysis on how much lidar data were used in the analysis (yearly and month-by-month)

Line 1 Pag. 3 Few minutes is not acceptable scientifically. AERONET specifications are available at NASA GSFC website.

Line 25 Pag.3 AERONET aerosol optical depth at 440nm should be greater than $\sim$0.05 since the calculation of Angstrom exponent at very low optical depths could introduce error due to the uncertainty of the AOD measurements (0.01) for wavelengths greater than 400nm. For high AOD and fine mode particles, the UV wavelengths may not fit on the logarithmic linear scale so some error can be introduced. How the authors dealt with those aspects?

Line 25 Pag 3. This is another potential serious issue underestimated and neglected in the paper. Why level 1.5 AERONET data are used? Level 1.5 data have pre-field calibration applied, however the calibration can change during the deployment (usually a linear rate due to slow deposition on the sensor head lenses), hence, the need for a post-field calibration. This means that Level 1.5 may show a large bias.

Line 1 Pag. 4 Pre-processing, not prepossessing

Line 25 Pag. 4 Why is not reported the used Lidar Ratio in the retrieval?

Line 30 Pag. 4 see Major Flaws section

Line 15 Pag. 5 why less structured? Is it due to the smoothing? If yes, how the profiles were smoothed?

Line 1 Pag. 6 The statement is not correct. The maximum height is reached not at noon (too generic) but at 12 Local Solar Time.

Line 15 Pag.8 The two distributions would not be similar if the lidar instrument reached full overlap closer to the ground.

Line 1 Page 9. "This is..." please make the sentence clearer.

Section 4.2.2 Integrated backscatter. It seems that this section doesn't make any sense. There is not added value in this intercomparison. It is exactly the same of integrating the aerosol extinction coefficient to retrieve AOD. Moreover, dividing arbitrary AERONET measurements by 50sr lidar ratio introduces very high errors.

While the reviewer recognized the potential importance and relevance of the comparison, the results reported in the paper are affected by severe methodological problems, which completely compromise their quality. The analysis of the present dataset should be reformulated removing all major methodological problems illustrated above.

The reviewer is available and willing to review again a completely revised version of the paper with consistent results obtained after addressing all the methodological problems

indicated above.

---

## Author Comment (AC1) · 5 Jun 2018

We would like to thank the reviewer for his/her fruitful comments that helped to improve the manuscript.

The subject of the manuscript is relevant to the journal, as different end-users need vertically-resolved aerosol profiles obtained from climatological observations, instead of using models (i.e. ARMA model). Nevertheless the paper shows conceptual errors that introduce serious issues that makes it not suitable for publication. Please refer to comment section for details.

General comments
After taking into account the feedback from all the reviewers we decided to proceed to the following major changes in the revised version of the manuscript.
– The Version 3, level 2.0 AERONET products replaced Version 2, level 1.5 products since they were recently published. When using Version 2, we preferred level 1.5 products because the AERONET timeseries was longer, starting at 2003. We noticed, however, that data in the period 2003-2005 that used to be categorized as Level 1.5 in Version 2 now are flagged as Level 2.0. Consequently, we decided to switch to level 2.0.
– The backscatter coefficient profiles and their respective columnar products (INTB) have been removed from Figure 3, Figure 4 and section 4. We deemed that these products were not providing any significant additional information and the comparison of the sunphotometer AOD at 355nm with the lidar INTB at 355nm caused unnecessary confusion.
– The aerosol optical properties analysis is now performed using solely night-time measurements. Since the backscatter products have been excluded, this mainly affects BAE355-532. We preferred this approach in order to improve homogeneity as the lidar ratio, a night-time product, is usually discussed hand-by-hand with BAE in the manuscript.
– A new paragraph that addresses sampling and consistency issues between the lidar and sunphotometer AOD at 355nm timeseries has been added. A number of tests has been performed in order to quantify the systematic biases that arise due to day/night differences and the fact that the lidar profiles typically start above 0.6km even if an overlap function is applied. The impact of the much lower resolution of the lidar sampling is also investigated.
– While re-processing the data, we detected and corrected some bugs that mainly affected the detection of the extreme values, the common boundaries of the two timeseries for the trend analysis and how the Mann-Kendal test had been applied. All the tables, figures and numeric values have been updated accordingly.

Major Flaws:
2) In the text it is clearly specified that a mean value for the lidar optical thickness is obtained by averaging measurements based on the elastic scattering technique with those obtained with the Raman scattering technique. The use of different techniques introduces a further bias.

The aerosol optical depth (AOD) at 355nm and the lidar ratio at 355nm are produced solely from night-time Raman measurements. Indeed the integrated backscatter (INTB) at 355nm, the INTB at 532 and the backscatter-related angstrom exponent BAE355-532 products were obtained both from daytime (Klett) and night time (Raman at 355nm and Klett at 532nm) backscatter profiles. Following the

reviewers' suggestions, we decided to remove the comparison of the annual cycles of those products with the sunphotometer cycles. The INTB plots (former figures 3c, 3d) have been removed and the BAE355-532 (former figure 3f) is no more compared with the sunphotometer angstrom 440-675. The BAE355-532 product is now obtained solely from night-time measurements. (See General comments above)

4) Even for daytime profiles, in the manuscript it is not even specified if an overlap correction is performed (i.e. shooting the lidar horizontally) and what is the extent of the lidar blind region and what the authors did to overcome this problem.

The reviewer is right. Indeed this was not clear in the manuscript. The overlap function is not applied for daytime measurements. For night-time measurements we apply the method of Wandinger et al. 2002. It allows the calculation of the lidar system's overlap function from Raman measurements. The correction is applied individually to each Raman measurement. It is limited to overlap values above 0.7 (Amiridis et al. 2005) and therefore cannot be extended down to the ground. A new paragraph (section 2.2) devoted to the system's overlap has been included in the manuscript. As mentioned in the general comments, the daytime measurements have been removed from the optical properties analysis.

1) The two climatologies cannot be compared and no conclusion can be drawn. AERONET is a daytime measurement, while lidar observations are taken and averaged independently, both during daytime and night-time. For sure, being different at night and day both the atmospheric conditions and aerosol emission sources (e.g. traffic and or household heating), a non-negligible bias is introduced in the analysis and consequently it is not possible to establish whether the correlation is good or not.
3) The most important contribution to the aerosol backscattering and extinction coefficients is coming from the first hundred meters that are heavily affected by overlap function. Only marginally In section 3.2, line 14 pag. 6 overlap problems are described . As night-time and daytime profiles are averaged together, an additional source of bias is introduced: what about the profiles for which the aerosol load is confined below 500m? It looks like those profiles cannot be compared at all with AERONET retrievals as in fact only a portion of those aerosol layer is detected.
5) It seems that the comparison has been performed based on data from the EARLINET database. In spite of points 1) and 3) above, the comparison has been done considering on average 52 days per year (corresponding to Monday morning schedule). On 52 these days, how many of them are cloud free? Are then the averages statistically meaningful? The paper is missing such analysis.

This study is not a direct comparison of the AOD 355nm values from the lidar and the sunphotometer. The consistency between the two climatologies is investigated by comparing annual cycles and long term trends. For this reason we did not originally perform a one by one comparison of the sunphotometer and lidar measurements. In order to investigate the possible effect of the sources of bias suggested by the reviewer to the annual cycle and trends we have isolated the common daily mean values between the two instruments and have performed the following diagnostics.  A new paragraph (section 4.5)  has been added in the manuscript concerning the findings mentioned below.

– Major flaw 5) suggests that the EARLINET sampling in combination with bad weather conditions could result to averages that are not representative and this would significantly affect the annual cycle and trends. We limited the AERONET dataset to only Monday and Thursday measurements to be compatible with the EARLINET schedule of night-time measurements. The resulting trend is -0.0090

per year, with a p-value at 0.000003 close to -0.0085 that occurs when using the whole dataset. The annual cycle seems stable with absolute differences smaller than 0.08 for every monthly average. To be on the safe side, we obtained the sunphotometer trend using only the daily means where both a sunphotometer and a lidar measurement were available. The resulting trend is -0.0089 per year, with a p-value at 0.035, still close to -0.0085 that occurs when using the whole dataset. Consequently, the lidar averages should be statistically meaningful and the uncertainty in the EARLINET trend should be less than +-0.0005 per year due to the limited sampling. Probably the length of the timeseries (14 years) compensates the sparse sampling rate. In the future, we plan to further analyze how the sampling and the timeseries length affect the climatological products produced from the columnar aerosol optical properties.

– Major flaws 1) and 3) suggest that, since the sunphotometer measurements are performed during the day and the lidar Raman measurements during the night, a systematic bias could be introduced. Additionally, the fact that, even after applying an overlap correction, our profiles seldom extend below 0.6km, could also contribute to this systematic bias. This bias is expected to produce an offset and/or seasonal discrepancies between the two datasets. Furthermore, an artificial trend could also be introduced to the lidar timeseries if the bias is non-periodically time-dependent. Changes in the systems overlap within the timeseries could produce such an effect. In order to investigate the aforementioned issues we isolate the common daily averages between the two datasets to ensure that only the overlap issues and the day/night discrepancies would contribute to the bias. We have computed the AOD at 355nm biases by subtracting the sunphotometer daily mean AOD from the lidar daily mean AOD per case. The seasonal biases and the total bias are calculated with a methodology similar to the one applied to the lidar and sunphotometer measurements. Spring and autumn biases are close to zero with values at 0.03 and -0.01 respectively. The winter seasonal bias is -0.15 while the summer bias is 0.13. The total bias is close to zero, at -0.003. Consequently, there is a minor offset towards slightly lower lidar AODs between the two annual cycles and a systematic estimation of higher lidar AOD values in summer and lower lidar AOD values in winter. This behavior is already visible in the monthly annual cycles (figure 4a), especially for summer. As far as the long term trend analysis is considered, even if the sunphotometer and the lidar AOD exhibit different seasonal patterns, we don't expect the trend values to be much affected since the seasonality has been removed from each timeseries individually (see section 4.4). The trend could only be affected by a non-periodical time dependence in the bias. We examine such effects by calculating the trend of the seasonal bias after removing the bias seasonality. We estimate a decreasing AOD355 trend of -00024 per year. A Mann-Kendal test is performed in order to check the significance of the this trend. It results to a p-value of 0.14 and therefore the trend hypothesis is rejected at the 5% acceptance interval. As a result, the long term trend of the lidar AOD should be free of systematic biases.

Specific Comments:
Line 2 Pag. 1 Measurements are not deployed, instruments are.
Line 3 Pag. 1 Please read: "These two instruments are members of two different networks. . ."
Line 4 Pag. 1 Please read:" The instruments are operated under a different time schedule."

The text has been modified to:
"For this purpose, measurements of two independent instruments, a lidar and a sunphotomer, were used. These two instruments represent two individual networks, the European Lidar Aerosol Network

(EARLINET) and the Aerosol Robotic Network (AERONET). They include different measurement schedules."

Line 12 Extinction is not defined. It is clear in the lidar community but for general audience it should be given a broader definition, as the vertical-resolved extinction coefficient

The text has been modified accordingly.

Line 16 Pag. 1 Please read:"a priori climatological profiles" Line 16 Pag. 1 Please read: " they can be used by modelers community"

The text has been modified to:
"This kind of information can be quite useful for applications that require a priori aerosol profiles. For instance, they can be utilized in models that require aerosol climatological data as input..."

The English in the abstract was improved. This should be extended on the whole manuscript. Often sentences are too long and convoluted.

Line 1 Pag. 2 atmospheric particles don't show variability, but concentrations or load yes.

The text has been modified to:
"The atmospheric aerosol load typically shows a significant spatial and temporal variability within the lower atmosphere."

Line 4 Pag.2 atmospheric conditions is more appropriate than wind circulation, other phenomena as convection are important.

The text has been modified to:
"Since the transportation is driven by the atmospheric conditions, ..."

Line 14 Pag.2: "The in situ technique. . ." please rephrase as the sentence is not clear.
The text has been modified to:
"In situ techniques focus on measurements of the aerosol properties close the ground. It is both challenging and costly to acquire those measurements in high altitudes (i.e. mounted on airplanes and unmanned aerial vehicles), especially on a routine basis."

Line 17 Pag. 2. References are not at all exhaustive. This comment is valid through all the manuscript.
More references have been included in the manuscript.

Line 31 Pag. 2 Raman indicates a person last name, then should be caps lock everywhere in the text.
The text has been modified accordingly.

Line 33 Pag. 2 As written before, it is missing an analysis on how much lidar data were used in the analysis (yearly and month-by-month)
A small paragraph has been included in the introduction with some information on the number of profiles that were used in the analysis.

Line 1 Pag. 3 Few minutes is not acceptable scientifically. AERONET specifications are available at NASA GSFC website.

The text has been modified to:
"Measurements are automatically performed every 15 minutes or less, depending on the sun's zenith angle (Holben et al. 1998, Dubovik et al. 2000)."

Line 25 Pag.3 AERONET aerosol optical depth at 440nm should be greater than ∼0.05 since the calculation of Angstrom exponent at very low optical depths could introduce error due to the uncertainty of the AOD measurements (0.01) for wavelengths greater than 400nm. For high AOD and fine mode particles, the UV wavelengths may not fit on the logarithmic linear scale so some error can be introduced. How the authors dealt with those aspects?

In order to investigate such errors we isolated the AERONET measurements where both the 340nm and the 440nm channels were available. Indeed, it seems that the conversion using the Angstrom at 440-675 leads to a systematic overestimation of the systematic overestimation to the extrapolated AOD340. In order to overcome this issue we apply a 2$^{nd}$ order polynomial fit to the logarithm of the AOD at 440nm, 675nm, 870nm and 1020nm (Soni et al. 2011). In the new figure 1, the extrapolated AOD340 from the polynomial seems in better agreement with the measured AOD340 than the constant-angstrom extrapolated AOD340. The polynomial approach is equivalent to applying an angstrom exponent with a linear spectral dependence. Using a constant angstrom (previous approach) is equivalent to assuming a linear fit in the logarithmic AOD. In the new version we extrapolate the AOD at 355nm from the polynomial. The error between the extrapolated and the measured AOD340 is within +-0.035 for 90% of the cases. The AOD440 uncertainty should be approximately +-0.02 and even higher for the UV (Kazadzis et al. 2016). Consequently, the new conversion ensures that the error introduced by the AOD extrapolation is typically close to the sun-photometer uncertainty. We also made sure that cases with an AOD355 < 0.05 are removed from the comparison. A new paragraph with details on this technique has been added in section 3.

Line 25 Pag 3. This is another potential serious issue underestimated and neglected in the paper. Why level 1.5 AERONET data are used? Level 1.5 data have pre-field calibration applied, however the calibration can change during the deployment (usually a linear rate due to slow deposition on the sensor head lenses), hence, the need for a post-field calibration. This means that Level 1.5 may show a large bias.

Taking into account the reviewer's suggestions, we have switched to Version 3 level 2.0 products in the revised version of the manuscript (see general comments above).

Line 1 Pag. 4 Pre-processing, not prepossessing
The text has been modified accordingly.

Line 25 Pag. 4 Why is not reported the used Lidar Ratio in the retrieval?
We use a different value depending on the availability of the closest in time Raman measurement and if not available, the lidar ratio of the dominant aerosol type is applied (Boeckman et al. 2004).

Line 30 Pag. 4 see Major Flaws section
The text has been modified accordingly.

Line 15 Pag. 5 why less structured? Is it due to the smoothing? If yes, how the profiles were smoothed?
The raman extinction profiles derive from the inelastic (387nm) range-corrected signal derivative. In order to calculate a derivative in a non-analytic way, information of nearby data points is required. This inevitably leads to "smoothing". We use a least squares fit approach (Papalardo et al. 2004) with a height dependent window of 300m below 2km, 600m between 2 and 4km and 900m above 4km.

Line 1 Pag. 6 The statement is not correct. The maximum height is reached not at noon (too generic) but at 12 Local Solar Time.
The text has been modified accordingly.

Line 15 Pag.8 The two distributions would not be similar if the lidar instrument reached full overlap closer to the ground.

Probably yes, the nocturnal stable boundary layer (SBL) would be also visible close to the ground. Despite that, the histogram provides evidence that the nocturnal residual layer top is present and quite similar to the daytime maximum PBL top. We have already mentioned (section 3.2.1) that the term "night-time PBL" would substitute the term "residual layer" for simplicity. We have added the following short description of the SBL in section 3.2.1 where we clarify that it is undetectable with the current setup.

Line 1 Page 9. "This is. . ." please make the sentence clearer.
The text has been modified to:
"The results of the columnar optical products and the geometrical products are displayed in monthly boxplots (figure 4) while the results of the profile optical products are exhibited in the form of seasonal average profiles (see section 4.3)."

Section 4.2.2 Integrated backscatter.
It seems that this section doesn't make any sense. There is not added value in this intercomparison. It is exactly the same of integrating the aerosol extinction coefficient to retrieve AOD. Moreover, dividing arbitrary AERONET measurements by 50sr lidar ratio introduces very high errors.

The reviewer is right. We deemed that this paragraph is not providing any significant additional information and the conversion of the sunphotometer AOD complicates the analysis. The section has been removed from the manuscript (see general comments above).

While the reviewer recognized the potential importance and relevance of the comparison, the results reported in the paper are affected by severe methodological problems, which completely compromise their quality. The analysis of the present dataset should be reformulated removing all major methodological problems illustrated above. The reviewer is available and willing to review again a completely revised version of the paper with consistent results obtained after addressing all the methodological problems

[revised manuscript text omitted]

---

## Author Comment (AC2) · 5 Jun 2018

We would like to thank the reviewer for his/her fruitful comments that helped to improve the manuscript.

In this study the authors present results from the climatological behavior of the aerosol optical properties over Thessaloniki during the years 2003-2017. Two independent datasets, representing two individual networks, the EARLINET and the AERONET, were applied to investigate the consistency and the statistical significance between both networks using geometrical and optical properties of aerosols. The analysis show a decreasing on AOD at 355 nm trends of -21.0% and -16.6% per decade for the EARLINET and the AERONET, respectively. Also, results show the dominance of dust and biomass burning on the free troposphere during summer. Different from other studies that considered only short time periods such as four or six years, and only one single kind of instruments (Lidar Raman), this study presented very important results of climatological studies of 14 years using two well establish networks. Overall, the manuscript is well well-organized and clearly presented. I'd like to suggest the acceptance of this manuscript after some revisions.

General comments
After taking into account the feedback from all the reviewers we decided to proceed to the following major changes in the revised version of the manuscript.
– The Version 3, level 2.0 AERONET products replaced Version 2, level 1.5 products since they were recently published. When using Version 2, we preferred level 1.5 products because the AERONET timeseries was longer, starting at 2003. We noticed, however, that data in the period 2003-2005 that used to be categorized as Level 1.5 in Version 2 now are flagged as Level 2.0. Consequently, we decided to switch to level 2.0.
– The backscatter coefficient profiles and their respective columnar products (INTB) have been removed from Figure 3, Figure 4 and section 4. We deemed that these products were not providing any significant additional information and the comparison of the sunphotometer AOD at 355nm with the lidar INTB at 355nm caused unnecessary confusion. The section's text has been modified accordingly.
– The aerosol optical properties analysis is now performed using solely night-time measurements. Since the backscatter products have been excluded, this mainly affects BAE355-532. We preferred this approach in order to improve homogeneity as the lidar ratio, a night-time product, is usually discussed hand-by-hand with BAE in the manuscript.
– A new paragraph that addresses sampling and consistency issues between the lidar and sunphotometer AOD at 355nm timeseries has been added. A number of tests has been performed in order to quantify the systematic biases that arise due to day/night differences and the fact that the lidar profiles typically start above 0.6km even if an overlap function is applied. The impact of the much lower resolution of the lidar sampling is also investigated.
– While re-processing the data, we detected and corrected some bugs that mainly affected the detection of the extreme values, the common boundaries of the two timeseries for the trend analysis and how the Mann-Kendal test had been applied. All the tables, figures and numeric values have been updated accordingly.

Section 2.1 The Lidar setup – page 3 – lines 16 to 19.
The authors use the Lidar data set between 2003 to 2017 and states, "since a long timeseries of data was necessary, only the extinction 355nm and the backscatter 355nm and 532nm products were included in the analysis. The dataset included in this study covers the period 2003-2017 in order to be chronologically consistent with the sunphotometer dataset."

The Lidar dataset used is from 2003 to 2017. It is well known that EARLINET has a well established standard pattern of quality assurance tests such as dark current, bin-shift, zero bin, trigger delay corrections, Telecover tests, Rayleigh fit, etc. Since when these tests are applied to Thessaloniki EARLINET station? Since 2003? What is the influence of these tests on the results of your comparisons? What type of errors or uncertainties the lack of these tests for the early dataset can take into account?

These test are currently incorporated in EARLINET's quality assurance internal checkups. Their main purpose is to report the status and monitor the stability of the system within the network. Submitting those tests once per year is mandatory for all the stations since 2012. However, most of these tests have been routinely performed in individual stations even before 2012. The dark current, bin-shift, trigger delay and Rayleigh fit test have been performed since 2001 in the station of Thessaloniki. The Rayleigh fit test, an essential diagnostic, allows us to determine if the lidar beam is aligned with the telescope axis. It is performed each time a measurement is processed. The dark current test has been typically performed before each measurement. The telecover test has been introduced in 2008 (Freudenthaler et al. 2008). It is a diagnostic tool used to determine the full overlap height of the each system channel with accuracy. It reveals more information on the origin of the overlap effect but it is not applied as a correction to the lidar products. The method of Wandinger at al. 2002 has been applied to our data since 2001 in order to determine the overlap function and consequently the full overlap height per Raman case. The following sentence has been added in section 3.1.

"Additionally, some quality standards have been established, in order to make the lidar products of the different systems comparable and to be able to provide quality-assured data sets of network products Freudenthaler et al. 2018."

Section 2.2 The sunphotometer - page 3 – lines 25 to 26
"The level 1.5 aerosol optical depth values (AOD) at 440nm and the angstrom exponent 440-670 during the period 2003-2017 were used in this study. The AOD at 440nm is preferred for the comparison with the lidar UV products in order to take advantage of the longer time series since the 340nm and 380nm channels were added in 2005."

Why not to use Level 2.0 data? What would be the differences on the trend results using the level 2.0 since it is quality assured; the final post-deployment calibration values are applied to the data set, and the aerosol optical depth data are inspected for possible cloud contaminated outliers. For AERONET level 1.5 data, when Angstrom parameter computed using all available channels between 440 and 870 nm is greater than -0.1 the point is considered cloud and pointing error free. Is the Level 1.5 AERONET data used for this study filtered using this assumption?

Taking into account the reviewer's suggestions, we have switched to Version 3 level 2.0 products in the revised version of the manuscript (see general comments above).

Section 2.2 The sunphotometer - page 3 – lines 26 to 28
"The AOD at 440nm is preferred for the comparison with the lidar UV products in order to take advantage of the longer time series since the 340nm and 380nm channels were added in 2005."

You add 2 year more on your climatology (2003 and 2004, since the 340nm and 380nm channels were added in 2005). How is the difference in your result considering these 2 years more?

We have overcome this issue by using the Version 3 AERONET products (see previous comment).

Subsection 4.2.1 - Aerosol Optical Depth – page 9 – 20 to 23
"The AOD cycle in the PBL and in the FT is presented in figure 3b. The contribution from the free troposphere seems to be comparable and even higher than the PBL contribution during April and the summer months. This is probably attributed to transported biomass burning aerosol during summer and spring in the FT (see section 4.2.2.4) The other months, especially March, exhibit a lower FT contribution."

It is possible to obtain some result or correlation of the biomass burning aerosol transported on the free troposphere using only AERONET AOD values?

There are techniques that allow the aerosol typing based on sunphotometer measurement (Hamill et al., 2016). However, they are not yet integrated in our routine processing algorithms. We plan to separate the desert dust and the biomass burning cases for both datasets in the future in order to analyze separately the long term trends of the transported aerosol cases.

Or considering the annual cycle of the monthly mean columnar products of AOD at 355nm in the whole column presented on figure 3 (a), is possible that AERONET is missing any aerosol layer on the free troposphere? How could it affect the results of the decreasing trends?

Indeed, since the sunphotometer measurements are performed during the day and the lidar Raman measurements during the night, a systematic bias could be introduced. Additionally, the fact that, even after applying an overlap correction, our profiles seldom extend below 0.6km, could also contribute to this systematic bias. This bias is expected to produce an offset and/or seasonal discrepancies between the two datasets. Furthermore, an artificial trend could also be introduced to the lidar timeseries if the bias is non-periodically time-dependent. Changes in the systems overlap within the timeseries could produce such an effect. In order to investigate the aforementioned issues we isolate the common daily averages between the two datasets to ensure that only the overlap issues and the day/night discrepancies would contribute to the bias. We have computed the AOD at 355nm biases by subtracting the sunphotometer daily mean AOD from the lidar daily mean AOD per case. The seasonal biases and the total bias are calculated with a methodology similar to the one applied to the lidar and sunphotometer measurements. Spring and autumn biases are close to zero with values at 0.03 and -0.01 respectively. The winter seasonal bias is -0.15 while the summer bias is 0.13. The total bias is close to zero, at -0.003. Consequently, there is a minor offset towards slightly lower lidar AODs between the two annual cycles and a systematic estimation of higher lidar AOD values in summer and lower lidar AOD values in winter. This behavior is already visible in the monthly annual cycles (figure 4a), especially for summer. As far as the long term trend analysis is considered, even if the sunphotometer and the lidar AOD exhibit different seasonal patterns, we don't expect the trend values to be much affected since the seasonality has been removed from each timeseries individually (see section 4.4). The trend could only be affected by a non-periodical time dependence in the bias. We examine such effects by calculating the trend of the seasonal bias after removing the bias seasonality. We estimate a

Subsection 4.2.2 Integrated Backscatter – page 9 – lines 25 to 27
"Another columnar optical product, the integrated backscatter (INTB) at 355nm and at 532nm, is presented in figure 3c and 3d. The AERONET equivalent is calculated by dividing the AOD at 355nm and at 532nm with a constant lidar ratio of 50 sr and it is also included in the figures."

What kind of error uncertainties and/or bias the authors could expect using the fixed Lidar ratio of 50 sr to calculate in INTB for the AERONET data? Since the the lidar ratio at 355 nm ranging from 45 to 70 sr according to statement on lines 2 and 3 of page 10, why not to use a mean fixed lidar ratio of 57 or 58 sr to calculate the AERONET integrated backscatter?

Indeed, in order to present the AERONET AOD355 together with the EARLINET INTB355 an assumption of a constant lidar ratio is required. Initially, we were more interested in comparing the general shape of the two annual cycles. The major benefit using the INTB355 could provide over using the AOD355 is that more EARLINET cases would be included (daytime measurements). Despite that, we decided that this is confusing and not really necessary for the study. Consequently, the Subsection 4.2.2 and Figure 3 (c, d) have been excluded from the manuscript (see general comments above).

One thing that is not clear on the manuscript is the consideration about the column AOD comparison between AERONET data, that performs measurements during daytime, and the AOD from Raman Lidar measured during the nighttime. What kind of correction or assumption the authors take into account for these cases?

The reviewer is right. We have included a new paragraph (section 4.5) in the revised manuscript where such issues are discussed (see the general comments and the previous response concerning the biases)

[revised manuscript text omitted]

---

## Author Response (AR2)

**Co-Editor Decision: Publish subject to minor revisions (review by editor)** (28 Jun 2018)

Line 9 Pag.3 Please read "lidar measurements were taken"
The text was modified accordingly.

Line 32 Pag. 5 Please remove the question mark
The text was modified accordingly.

Line 9 Pag. 6 Please read: "At night, the detection of the nitrogen vibrational Raman channels (387 and 607 nm) has a sufficiently high signal-to-noise ratio to compute the inversion"
The text was modified accordingly.

Line 30 Pag. 6 Please read "the longer wavelengths, less sensitive to Rayleigh scattering..."
The text was modified accordingly.

Line 5 Pag 7 I would also the works of Haeffelin et al., 2012 (https://doi.org/10.1007/s10546-011-9643-z) and Milroy et al 2012 (http://dx.doi.org/10.1155/2012/929080)
The two citations have been included.

Line 8 Pag. 10 Please read: "since is too close to the ground"
The text was modified accordingly.

Line 10 Pag. 10 Please read "it is important..."
The text was modified accordingly.

Paragraph 4.2.1 It is scientifically relevant to quantify the correlation of EARLINET and AERONET averages (Fig 4a) using Pearson and CC cf. Lolli et al., 2013 (https://doi.org/10.5194/amt-6-3349-2013)
The text was modified accordingly.

Line 32 Pag 19. The reference shows first names instead of last names
The reference was corrected.

[revised manuscript text omitted]